# ACTION ABSTRACTIONS FOR AMORTIZED SAMPLING

**Oussama Boussif**[1,2]\*  **Léna Néhale Ezzine**[1,2]  **Joseph Viviano**[1]  **Michał Koziarski**[1,3]
**Moksh Jain**[1,2]  **Nikolay Malkin**[5]  **Emmanuel Bengio**[4]  **Rim Assouel**[1,2]
**Yoshua Bengio**[1,2,6]

[1]Mila – Québec AI Institute  [2]Université de Montréal  [3]University of Toronto
[4]Valence Labs, Recursion  [5]University of Edinburgh  [6]CIFAR AI Chair

## ABSTRACT

As trajectories sampled by policies used by reinforcement learning (RL) and generative flow networks (GFlowNets) grow longer, credit assignment and exploration become more challenging, and the long planning horizon hinders mode discovery and generalization. The challenge is particularly pronounced in entropy-seeking RL methods, such as generative flow networks, where the agent must learn to sample from a structured distribution and discover multiple high-reward states, each of which take many steps to reach. To tackle this challenge, we propose an approach to incorporate the discovery of action abstractions, or high-level actions, into the policy optimization process. Our approach involves iteratively extracting action subsequences commonly used across many high-reward trajectories and 'chunking' them into a single action that is added to the action space. In empirical evaluation on synthetic and real-world environments, our approach demonstrates improved sample efficiency performance in discovering diverse high-reward objects, especially on harder exploration problems. We also observe that the abstracted high-order actions are potentially interpretable, capturing the latent structure of the reward landscape of the action space. This work provides a cognitively motivated approach to action abstraction in RL and is the first demonstration of hierarchical planning in amortized sequential sampling. Code is available at `https://github.com/GFNOrg/Chunk-GFN`.

## 1 INTRODUCTION

Reinforcement learning (RL) relies on a stochastic policy $\pi_\theta$ to generate potentially long trajectories $\tau$ of actions $\alpha$ to obtain a reward $r$. Standard RL methods take learning steps on the policy parameters $\theta$ using a loss function that reinforces actions which maximize reward (Sutton & Barto, 2018). In the case of diversity-seeking RL methods, such as generative flow networks (GFlowNets; Bengio et al., 2021; 2023) – a special case of entropy-regularized RL (Ziebart et al. (2008); see Tiapkin et al. (2023); Deleu et al. (2024)) – the loss function may instead encourage the policy to sample terminal states with probability proportional to their reward. In both cases, longer trajectories make training more difficult due to the problem of *credit assignment*, *i.e.*, the propagation of a learning signal over long time horizons (Sutton & Barto, 2018).

The difficulty of credit assignment grows with trajectory length: it has been shown that temporal difference learning methods require an exponential (in the trajectory length) number of updates to correct learning bias, while Monte Carlo methods see the number of states affected by delayed rewards grow exponentially with the number of delay steps (Arjona-Medina et al., 2019). Previous work addressed the challenge of credit assignment through long trajectories by propagating a learning signal to specific moments in the trajectory, skipping over intermediate actions and effectively reducing the trajectory length (Ke et al., 2018; Liu et al., 2019; Hung et al., 2019; Sun et al., 2023).

Recent work has shown that trajectory compression improves credit assignment in RL. For instance, Ramesh et al. (2024) show that it is beneficial to explicitly shorten trajectories by dropping the intermediate states which are highly predictable given their predecessors, related to the principle of history compression (Schmidhuber, 1992). Other recent work used a similar compressibility criterion directly as a reward term: by encouraging the policy to implicitly learn trajectories that could be easily compressed or predicted (*e.g.*, that contain repeating elements), the RL agents achieve better

---

\*Correspondence to: `oussama.boussif@mila.quebec`

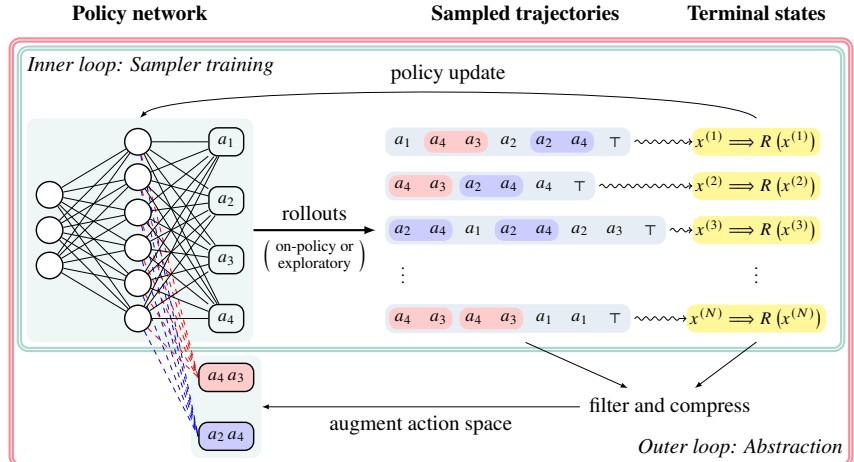

Figure 1: **Chunking procedure**. Starting with a policy, we generate trajectories consisting of action sequences. The trajectories are then filtered to retain only the high-reward samples, which are passed to a tokenizer. The tokenizer identifies frequently occurring "chunks" which are added to the action space. The process is repeated till convergence.

sample efficiency (Saanum et al., 2023). These works suggest that the lower description length of action sequences confers some benefit in terms of the credit assignment challenge.

Analogies exist between compression in artificial and natural learning systems. A long line of research in psychology investigates "chunking", a phenomenon whereby humans tend to record smaller units of information as part of larger high-order units. Chunking is believed to reduce the load on working memory when reasoning about complex problems which would take too many bits to encode using only lower-level units (Thalmann et al., 2019; Gobet et al., 2001; Miller, 1956; Johnson, 1970).

In RL, *options and macro-actions* generalize primitive actions and allow for temporally extended sequences of actions, or temporal abstractions, that facilitate more efficient exploration and enable hierarchical learning. Macro-actions have inconsistently been found to speed-up learning in RL and enhance credit assignment (Randlov, 1998; Sutton et al., 1999).

Drawing on the above work, we propose to abstract high-order "chunks" *online* from the trajectories sampled by a GFlowNet or RL policy, producing an action space that grows during training. In contrast to (Saanum et al., 2023), compressibility is not enforced through objectives – rather, we harness the virtuous cycle of emergent compressibility of the policy's trajectories and the accelerated learning that results from shorter action sequences. In addition, we find that these high-order actions accurately recover the latent structure of the underlying distribution.

Our contributions can be summarized as follows:

- We present ACTIONPIECE, a method for extracting high-order actions, or "chunks," from sampled trajectories using tokenization techniques. ACTIONPIECE includes two variants: ACTIONPIECE-INCREMENT and ACTIONPIECE-REPLACE. Both approaches are compatible with any sampler at no significant cost.

- We introduce *ShortParse*, a new backward policy for the GFlowNet for sampling short trajectories.

- We evaluate our approach on multiple environments and determine that it accelerates mode discovery, improves density estimation, and reduces description length of the samples.

- We demonstrate that learned chunks represent the latent structure of the target distribution and are transferable to different algorithms and new tasks.

## 2 RELATED WORK

**Macro-actions** are a sequence of actions assembled from an atomic action set (Randlov, 1998), a form of temporal abstraction, and have been the focus of a long line of work. The idea stems from early proposals of automatic induction of *macros* for programs using various rules or heuristics (Iba, 1989; Laird et al., 1986); see Sutton et al. (1999) on the related framework of *options* and a review of macro-actions. In our context, a macro-action is a policy representing a series of atomic

actions to be executed at once, which in some cases has been shown to reduce the time required to discover the optimal policy and action-value function in the RL context (McGovern & Sutton, 1998; Laird et al., 1986). Previous work has proposed composing macro-actions from n-grams to speed up search during planning (Dulac et al., 2013) and employing them to construct form of a hierarchy of policies with different temporal resolutions (Precup & Sutton, 1997; Dayan & Hinton, 1992). Such methods have produced mixed results, either helping or hindering performance depending on the appropriateness of the macro-actions to the task (McGovern et al., 1997; Jong et al., 2008). In cases where macro-actions were helpful, their utility was explained by improved credit assignment and exploration. Previous work has also used evolutionary algorithms for discovering macro-actions, which was an effective approach on common Atari games (Chang et al., 2022). Others proposed to learn macro-actions as either actions that are repeated a number of times, or commonly-occurring sub-sequences of a fixed length (Durugkar et al., 2016) while training a deep RL policy. In this work, speed of convergence is the most reliable improvement from the introduction of macro-actions, but variance of returns is also decreased, and in some specific settings returns improved. The settings where macro-actions are useful appear to contain more shared structure between the learned macros and the latent structure of the environment and reward landscape.

Work on **library learning** in program induction (*e.g.*, Ellis et al. (2018); Nye et al. (2020); Chaudhuri et al. (2021)) has often sought to incrementally modify the domain-specific language of interest so as to reduce the complexity of useful programs, following the minimum description length principle (Rissanen, 1978). DreamCoder and related approaches (Ellis et al., 2021; Claire et al., 2023; Ellis et al., 2018; Bowers et al., 2023) accomplish this by introducing new subroutines during a distinct abstraction phase during training. These algorithms leverage a *library* of solutions to given problems which serve as a prior (see also Tian et al. (2020); Liang et al. (2010)). Many other works in this space only *implicitly* learn a library in the form of a policy which emits a program when conditioned on a program (*e.g.*, Ellis et al. (2019); Devlin et al. (2017)). Similar works exist in the related domain of proof synthesis (*e.g.*, Vyskočil et al., 2010; Zhou et al., 2024).

**Credit assignment** is a fundamental problem in sequential decision-making, known to arise from a few properties of the Markov decision process (MDP) in which the learner acts: its depth (the typical number of steps between the initiation of a trajectory and a reward); its density (the number of nonzero rewards in the typical trajectory) and its breadth (the number of possible routes from the initial state to a reward; for a review, see Pignatelli et al. (2024). In RL, Hindsight Credit Assignment (HCA; Harutyunyan et al., 2019) introduces a new family of algorithms where credit is assigned to past decisions based on the likelihood of them having led to the observed outcome. In GFlowNets, the trajectory balance objectives (Malkin et al., 2022; Madan et al., 2023), which consist of a loss depending on a whole action sequence rather than a single transition, and the forward-looking flow parametrization (Pan et al., 2023), which is effective when the energy can be decomposed additively into per-time-step components, are both motivated by the credit assignment challenge.

## 3 PRELIMINARIES

We begin by briefly introducing the concept of *amortized sampling* and summarize the learning problem solved by GFlowNets and the objective used, then introduce the two RL algorithms on which we also evaluate our action abstraction approach.

**Amortized sampling** refers to the process of sampling from a functional approximation of the target distribution, where the computational cost of iterative sampling is shifted to the model's optimization process. This allows for efficient and rapid sampling once training is complete.

Let $G = (S, A)$ be a directed acyclic graph. Nodes $s \in S$ are called *states* and edges $(s \rightarrow s') \in A$ are *actions*, which define the relations "$s$ is a parent of $s'$" and "$s'$ is a child of $s$". We assume that there is a unique state $s_0 \in S$ with no parents (*initial state*) and denote the set of states with no children (*terminal states*) by $X$. One interprets $X$ as the set of complete "objects" that one can sample and the actions as constructive steps that incrementally build an object.

A key assumption needed to formulate action abstraction is that there is a correspondence between the actions available at different states. This is a common assumption in RL – in MDPs, the action space is typically the same at all states, but some actions may be "masked" or "invalid" at some states – but is not a typical GFlowNet assumption. Here, we assume that there is a global set of actions $\mathcal{A}$ and that at each nonterminal state $s$ a subset $\mathcal{A}_s \subseteq \mathcal{A}$ of these actions is available. We fix a bijection between $\mathcal{A}_s$ and the set of children of $s$; thus, applying action $a \in \mathcal{A}_s$ at state $s$ transitions to the child state corresponding to $a$, denoted $s + a$. (For example, when constructing

sequences over some alphabet by adding one symbol at a time to the end, $\mathcal{A}$ may be the alphabet, with $\mathcal{A}_s = \mathcal{A}$ for all $s$.)

A *(forward) policy* $P_F$ is a collection of distributions over $\mathcal{A}_s$ (identified by the aforementioned bijection with the set of children of $s$) for every non-terminal state $s$. A forward policy induces a distribution over *complete trajectories*, or paths in the directed graph from $s_0$ to a terminal state $s_n \in X$:

$$P_F(\tau = (s_0 \xrightarrow{a_0} s_1 \xrightarrow{a_1} \dots \xrightarrow{a_{n-1}} s_n)) = \prod_{i=0}^{n-1} P_F(a_i \mid s_i), \tag{1}$$

where $s_i \xrightarrow{a_i} s_{i+1}$ indicates that $s_i + a_i = s_{i+1}$. The marginal likelihood of sampling a terminal state $x \in X$ is then

$$P_F^{\top}(x) = \sum_{\tau \rightsquigarrow x} P_F(\tau),$$

where the sum is over all complete trajectories ending in $x$. (Note that this sum may be intractable to compute exactly if the number of trajectories leading to $x$ is large.)

**Trajectory balance.** GFlowNet training aims to approximate $P_F$ (for example, parameterized as a neural network $P_F^{\theta}(\cdot \mid \cdot)$) so as to make $P_F^{\top}$ proportional to a given reward function $R : X \to \mathbb{R}^+$. Various objectives for achieving this exist; here, we use the *trajectory balance* (TB) objective (Malkin et al., 2022). TB requires introducing two auxiliary objects: a learned scalar $Z_{\theta}$ (which estimates the normalizing constant $Z = \sum_{x \in X} R(x)$) and a (fixed or learned) *backward policy* $P_B$, which is a collection of distributions $P_B(\cdot \mid s)$ over the parents of every non-initial state $s$. Similarly to equation 1, $P_B$ induces a distribution over complete trajectories ending in $x$:

$$P_B(\tau = (s_0 \to \dots \to s_n = x) \mid x) = \prod_{i=0}^{n-1} P_B(s_i \mid s_{i+1}).$$

The TB objective aims to make $P_F(\tau)$ proportional to $P_B(\tau \mid x)R(x)$ for every trajectory $\tau$ ending in $x$, a constraint that implies that $P_F^{\top}(x) \propto R(x)$. The training loss for a trajectory $\tau$ enforces this proportionality:

$$\mathcal{L}(\tau; \theta) = \left( \log[Z_{\theta} \cdot P_F^{\theta}(\tau)] - \log[P_B^{\theta}(\tau \mid x) \cdot R(x)] \right)^2. \tag{2}$$

If $\mathcal{L}(\tau; \theta) = 0$ for all $\tau$, then the desired proportionality is satisfied and $P_F^{\top}(x) \propto R(x)$. The training procedure takes gradient steps with respect to $\theta$ to minimize $\mathcal{L}(\tau; \theta)$ over trajectories sampled from some behaviour policy $\pi(\tau)$. As we aim to simultaneously minimize equation 2 to 0 for all trajectories, the behaviour policy can be any full-support distribution, such as $P_F$ itself (*on-policy* training) or a distribution chosen to promote exploration (*off-policy* training).

**RL algorithms.** The above setting can be translated to RL terminology in a straightfoward manner: one defines an MDP with states $S$, action space $\mathcal{A}$, deterministic transition function $T(s, a) = s + a$, and reward $r(s \to s') = \log R(s')$ if $s' \in X$ and 0 otherwise. (The log is necessary for the equivalence of GFlowNet and RL algorithms in certain cases to hold; see Tiapkin et al. (2023); Deleu et al. (2024).) We now recall two RL objectives, restricting to this sparse-reward setting with discount factor 1.

The **Advantage Actor-Critic (A2C)** algorithm (Degris et al., 2012) combines policy-based and value-based methods to improve training stability and efficiency. Two networks are learned jointly: a policy network $P_F^{\theta}$, which plays the role of the actor, and a state-action value (or critic) network $Q_w^{\theta}$ to measure the 'goodness' of actions taken by the actor. The two objects are learned using the following update:

$$\theta \leftarrow \theta - \lambda_{\theta} Q_w^{\theta}(s_t, a_t) \nabla_{\theta} \log P_F^{\theta}(a_t \mid s_t)$$
$$w \leftarrow w - \lambda_w (r(s_t, a_t) + Q_w(s_{t+1}, a_{t+1}) - Q_w(s_t, a_t)) \nabla_w Q_w(s_t, a_t)$$

where $\lambda_{\theta}$ and $\lambda_w$ are learning rates. To further stabilize training, a *baseline* is usually used, and $Q(s_t, a_t)$ is replaced with $Q(s_t, a_t) - \mathbb{E}_{a_t \sim P_F^{\theta}}[Q(s_t, a_t)]$, reducing the variance of the gradient with respect to $w$ while keeping it unbiased.

The **Soft Actor-Critic (SAC)** algorithm (Haarnoja et al., 2018) is an off-policy algorithm that is trained to maximize a trade-off between the expected return and the entropy of the policy $P_F$, the

latter encouraging more exploration. Using the notation introduced above, SAC's objective is:

$$\arg\max_{P_F} \mathbb{E}_{\tau \sim P_F} \left[ \sum_{t=0}^{|\tau|-1} (r(s_t, a_t) + \alpha H(P_F(\cdot \mid s_t))) \right], \tag{3}$$

where $\alpha$ is an entropy regularization parameter. To satisfy this objective, we train a state-value function $Q_w$ and a policy network $P_F^\theta$ to satisfy :

$$Q_w(s_t, a_t) = r(s_t, a_t) + \mathbb{E}_{(s_{t+1}, a_{t+1}) \sim P_F^\theta} \left( Q_w(s_{t+1}, a_{t+1}) - \alpha \log P_F^\theta(.|s_{t+1}) \right) \quad \text{(Bellman equation)}$$

$$\theta = \arg\min_\theta \mathbb{E}_{s_t \sim q} D_{\text{KL}} \left( P_F^\theta(\cdot \mid s_t) \,\|\, \frac{\exp(Q_w(s_t, \cdot))}{Z_w(s_t)} \right) \quad \text{($q$ can be any off-policy distribution).}$$

## 4 DISCOVERING ACTION ABSTRACTIONS FOR AMORTIZED SAMPLERS

Training amortized samplers in discrete compositional spaces becomes increasingly difficult as the number of steps in trajectories grows. Consider sampling a graph one node at a time while connecting them to previous nodes. For example, a graph with $n$ takes $O(n^2)$ sampling steps. Sampling long strings autoregressively presents a similar challenge: sampling one character at a time makes the process difficult due to the vast search space of full sequences, which are predominantly low-likelihood samples. In the case of large language models, sampling subsequences (tokens) which commonly occur in the text corpora instead of individual characters reduces the trajectory length required to encode high-likelihood sentences. This has generally been found to decrease both training and inference costs (Wu, 2016; Kudo, 2018), and can be viewed a strategy for trading off reducing the depth of the MDP in exchange for increased breadth, which potentially makes the task easier to learn (Pignatelli et al., 2024).

Drawing inspiration from this observation, we propose a general approach for sampling discrete objects that infers actions that effectively shorten the trajectory length while producing samples from the target distribution. In particular, our approach consists of augmenting existing samplers with an action abstraction discovery step, a form of "chunking" as originally described by cognitive psychologists (Miller, 1956), which is applicable to any sampler, and consists of three major steps (see Algorithm 1 for the overall approach):

1. **Generating an *action corpus***: We first generate a set of $N$ action sequences from the sampler, optionally also drawing a portion $p$ of these from a replay buffer of previously-drawn high-reward trajectories.
2. **Chunking**: We apply a tokenization algorithm common in NLP, byte pair encoding (BPE; Gage, 1994), to the $N$ action sequences to obtain new tokens ("chunks") to be added to the action space.
3. **Augmenting the action space**: Finally, we add the new abstracted actions to the action space. Whenever the abstracted action is chosen, its constituent actions are executed in order[1].

---

**Algorithm 1** Training policies with chunking

---

1: **Initialize** $\mathcal{A}$ (action space), $\mathcal{B}$ (replay buffer), $M$ (maximum number of iterations), $p$ (proportion to sample from the replay buffer if it exists) and $N$ (number of trajectories), and $k$, (chunking frequency).
2: **for** $n = \{1, 2, \ldots, M\}$ **do**
3:      loss = train($\mathcal{A}, \mathcal{B}, n$)                               ▷ Train the amortized sampler
4:      **if** $n\%k == 0$ **then**
5:          Sample $\lfloor pN \rfloor$ from $\mathcal{B}$ to get action sequences $\{a_{\mathcal{B}}^i\}_{i=1}^{\lfloor pN \rfloor}$.
6:          Sample $N - \lfloor pN \rfloor$ from the sampler to get action sequences $\{a^i\}_{i=1}^{N-\lfloor pN \rfloor}$.
7:          $\mathcal{S} = \{a^i\}_{i=1}^{N-\lfloor pN \rfloor} \cup \{a_{\mathcal{B}}^i\}_{i=1}^{\lfloor pN \rfloor}$
8:          $C = \text{chunking}(\mathcal{S})$                       ▷ Generate novel chunks
9:          $\mathcal{A} \leftarrow \mathcal{A} \cup C$                         ▷ Update the action space
10:      **end if**
11: **end for**

---

[1] This differs from the options framework where the agent is allowed to only partially execute the option. If an abstracted action representing a sequence of actions is not possible to execute (*i.e.*, would lead to the application at some $s$ of an action not in $\mathcal{A}_s$), the abstracted action is masked by the policy.

This approach is similar to the offline method proposed by Zheng et al., which learns a static set of macro-actions from a dataset in the continuous control setting. In contrast, our work introduces a method where the library of macro-actions is constructed online and updated iteratively within the context of amortized sampling.

**Chunking mechanisms.** We consider the following two approaches, comparing them to the ATOMIC sampler (without action space compression):

- ACTIONPIECE-INCREMENT: We apply the tokenizer on the action corpus and add the most frequent token found to the action space, which grows by one element each time this is performed. This approach gradually builds up the action space but is susceptible to repetitions in chunks or low quality set of action sequences used for checking.
- ACTIONPIECE-REPLACE: This method instead takes the $M$ most frequent tokens found by the tokenizer from the action corpus (to be appended to the initial action space). Each time abstraction is performed, the entire action space might be replaced by a new one (aside from the atomic tokens). This effectively reduces the chance of storing redundant chunks and allows us to potentially discover many useful chunks in one step.

**Chunking frequency.** ACTIONPIECE require choosing "*when to chunk*" during training. A simple approach (used in most experiments) is to simply chunk every $k$ iterations. The alternative is to chunk based on some criterion being satisfied. In the case of GFlowNets, the TB loss can be used as a proxy for the "fit" of the sampler to the current action space (see Appendix B), and we therefore evaluated waiting for the loss to drop below a certain threshold before a round of chunking.

## 5 EXPERIMENTAL SETUP

We benchmark SAC, A2C and GFlowNet along with a random sampler baseline that samples uniformly at random a valid action at each step. For the GFlowNet, we use trajectory balance (Malkin et al., 2022) and include three different choices for the backward policy:

- **Uniform** $P_B$: This backward policy chooses backward *parents* uniformly at random.
- **MaxEnt** $P_B$: This backward policy chooses backward *trajectories* uniformly at random. Note that MaxEnt $P_B$ and Uniform $P_B$ are equivalent for Tree MDPs since there is only a single backward action at each step (Tiapkin et al., 2023; Mohammadpour et al., 2024).
- **ShortParse** $P_B$: We introduce a new fixed backward policy aimed at sampling the most compact backward trajectory in terms of the number of trajectory steps (see subsection B.6).

**A policy parametrization for nonstationary action spaces.** Dealing with a varying action space as a result of chunking requires a compatible parameterization of the policy to handle the evolving action space. Instead of a final layer that outputs logits for a fixed number of actions, the policy instead takes as input the current state and outputs an *action embedding* $\mathbf{q_t} \in \mathbb{R}^d$ where $d$ is the action embedding dimension (similar to Chandak et al., 2019). Now let $\mathcal{A} = \{a_i\}_{i=1}^{|\mathcal{A}|}$ where $|\mathcal{A}|$ is the number of actions in the action space $\mathcal{A}$. We use an action encoder $f_\theta(a_i) \in \mathbb{R}^d$ that computes an embedding for the actions $a_i$. We parametrize $f_\theta$ as an LSTM (Hochreiter & Schmidhuber, 1997) that takes the sequence of atomic actions making up a chunk as input (possibly of length 1 if the chunk is a single atomic action). Let $f_\theta(\mathcal{A}) \in \mathbb{R}^{|\mathcal{A}| \times d}$ denote the matrix of embeddings of all actions in the action space. The logits given by the policy network are:

$$\ell_t = \frac{f_\theta(\mathcal{A})\mathbf{q_t}}{\sqrt{d}} \in \mathbb{R}^{|\mathcal{A}|} \tag{4}$$

**Tasks.** To understand the impact of chunking, we consider a diverse set of environments consisting of three synthetic tasks and a practical task: bit sequence generation (Malkin et al., 2022), Fractal-Grid (adapted from the standard hypergrid; Bengio et al., 2021), a graph generation environment, and the practical task of RNA sequence generation (L14_RNA1; Sinai et al., 2020). We provide further details about each environment in Appendix B.

## 6 RESULTS

### 6.1 DOES CHUNKING IMPROVE GENERATIVE MODELING?

**Does chunking accelerate mode discovery?** Figure 2 shows the number of modes discovered throughout training as a function of the number of visited states. Note that we report the cumulative number of modes, *i.e.*, counting all modes sampled for a given number of visited states, and the difficulty of the task (in terms of the size of the MDP) increases from left to right. Across all

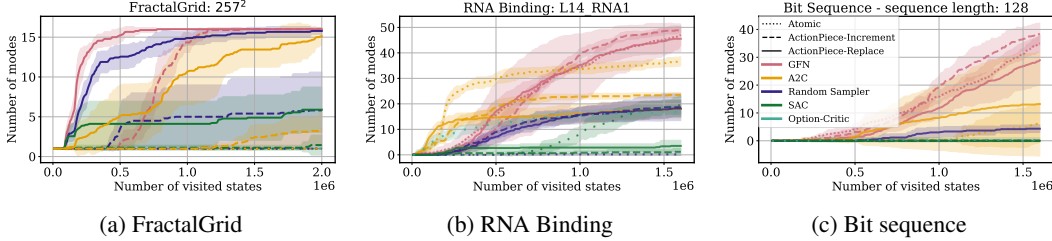

(a) FractalGrid        (b) RNA Binding        (c) Bit sequence

Figure 2: **Cumulative number of modes discovered during training**. Chunking helps across all environments, especially in FractalGrid where all samplers get stuck in the first mode but chunking unlocks exploratory abilities to fetch faraway modes. Shaded area represents the standard deviation across 6 seeds.

tasks, ACTIONPIECE outperforms the ATOMIC counterpart for the GFlowNet sampler in terms of the speed of mode discovery. Results for other algorithms are mixed. ACTIONPIECE helps A2C and SAC in FractalGrid, but hurts performance for RNA binding. Moreover ACTIONPIECE-REPLACE helps A2C in BitSequence, while ACTIONPIECE-INCREMENT hurts performance. The greedy behavior of both SAC and A2C provides a possible explanation: when a mode is discovered, it gets sampled more often resulting in the addition of a large chunk (*e.g.*, Figure 10,12). This chunk will result in high-reward samples and consequently chunks added beyond this will contain parts of this chunk hurting exploration. This is further supported by the the Option-Critic performance. This issue is mitigated in GFlowNets due to the exploratory behavior imparted by the learning objective. The random sampler with ACTIONPIECE achieves strong performance only on simple tasks, with a degradation in performance for harder problems, suggesting random sampling with action abstraction produces meaningful chunks only in cases where the problem is sufficiently simple.

**Does chunking improve density estimation?**
Next, we study the impact of chunking on density estimation in the context of GFlowNets, which learn a policy to sample from a target distribution. Figure 3a shows the trajectory balance loss for all chunking mechanisms over the 2D grid in the FractalGrid environment. This serves as a visualization for how well each state is correctly learned by the GFlowNet. Chunking, and particularly ACTIONPIECE-INCREMENT, positively rein-

Table 1: Comparison of chunking mechanisms based on various metrics for the graph environment with 7 nodes. We use three seeds with the standard deviation indicated in parentheses. We sample 10,000 graphs for computing the ELBO Gap and draw 40 backward trajectories for each graph to compute the JSD and L1 distance.

| Mechanism | ELBO Gap | JSD | L1 Distance |
|---|---|---|---|
| ATOMIC | $0.72_{\pm 0.39}$ | 0.38 | 1.44 |
| ACTIONPIECE-INCREMENT | $\mathbf{0.25_{\pm 0.14}}$ | **0.36** | **1.39** |
| ACTIONPIECE-REPLACE | $0.35_{\pm 0.11}$ | 0.40 | 1.47 |

forces the inherent exploration of GFlowNets (as seen in the previous section), helping enable the sampler to model the entire state space and sample accurately from the target distribution. This is further supported by the L1 distance, JSD, and ELBO Gap (see subsection B.7) during training illustrated in Figure 3b for both grid size $65^2$ and $129^2$ and all chunking mechanisms. Across all sizes of the task, chunking accelerates the training of the sampler. Figure 3c and 3d show Spearman's rank correlation coefficient between the target and the learned distribution for the L14_RNA1 and Bit Sequence tasks respectively. The correlation is computed for a range of reward thresholds spanning all samples to only those with a reward larger than 0.93 to evaluate how well the sampler captures the high reward regions. For L14_RNA1, ACTIONPIECE-INCREMENT performs on par with ATOMIC, and even slightly better for high-reward objects, whereas ACTIONPIECE-REPLACE lags behind. In the bit sequence task, however, chunking results in marginally worse performance. Finally, Table 1, shows that ACTIONPIECE-INCREMENT improves density estimation in the graph environment, while ACTIONPIECE-REPLACE is also competitive. With some exceptions, these results point towards chunking improving density estimation.

### 6.2 DO CHUNKS CAPTURE THE STRUCTURE IN THE UNDERLYING DISTRIBUTION?

In this section, we analyze the discovered chunks and study whether they capture some structure in the underlying distribution. For instance, RNA molecules are typically composed of building blocks (or chunks in our terminology) called codons that consist of sequences of three nucleotides. We use the RNA Binding (L14_RNA1) task for all the analyses in this section as we have access to a dataset of high-reward objects.

**Do chunks represent latent structure of the distribution?** Here, we dive into the structural relationship between chunks and the objects generated using the chunks. We consider a **Chunk Occurrence** metric where we compute the average number of times a chunk occurs in objects in the dataset, and a **Chunk Coverage** metric where we compute the number of objects in the dataset that contain a chunk, normalized over the total number of objects.

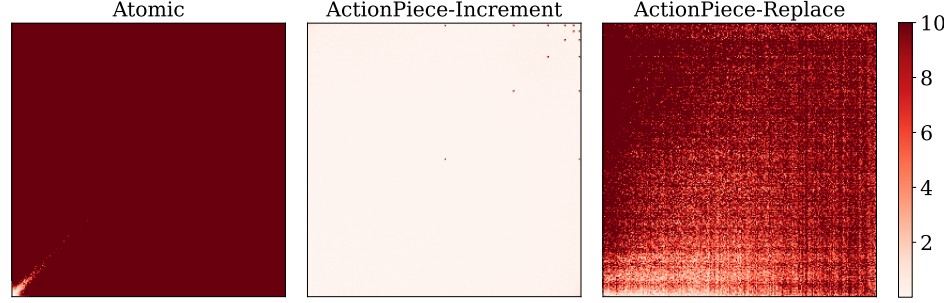

(a) TB loss for all chunking mechanisms in FractalGrid for a grid size of $257^2$. All the losses are clipped to a maximum value of 10 for ease of visualization.

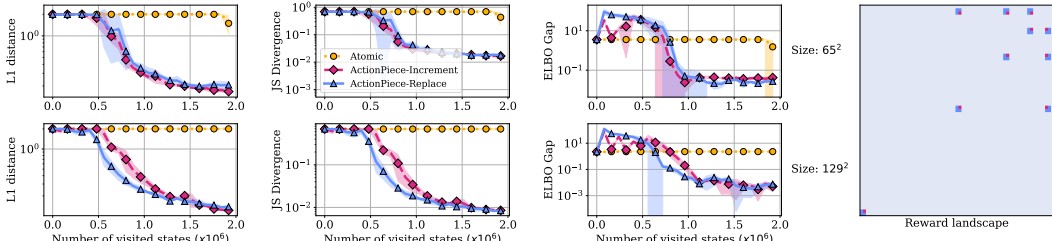

(b) The evolution of sampling metrics for all chunking mechanisms during training. The top row is for FractalGrid of size $65^2$ whereas the bottom row corresponds to that of size $129^2$. On the right, we plot the reward landscape for a grid of size $65^2$ for ease of visualization.

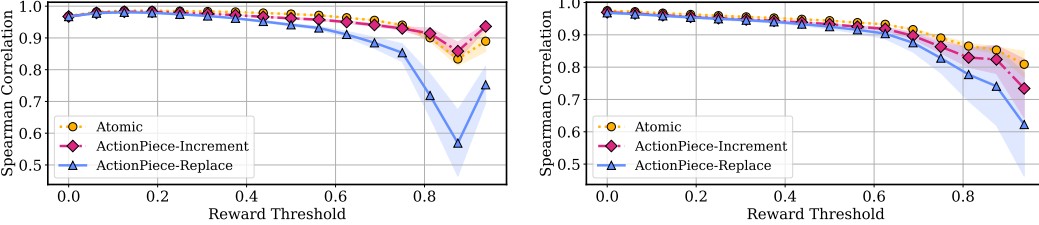

(c) Spearman correlation between the logreward and the learned log-likelihood for the L14_RNA1 task.

(d) Spearman correlation between the logreward and the learned log-likelihood for the bit sequence task.

Figure 3: **Effect of chunking on density estimation.** Analysis of the effect of different chunking mechanisms on density estimation in the FractalGrid, bit sequence, and RNA binding tasks.

Table 2: Chunk Occurrence and Coverage for different samplers and chunking mechanisms for the L14_RNA1 environment across 6 seeds.

| Sampler | Mechanism | Chunk Occurrence | | | Chunk Coverage | | |
|---|---|---|---|---|---|---|---|
| | | Mean | Median | SD | Mean | Median | SD |
| GFlowNet | ACTIONPIECE-INCREMENT | **1.11** | **1.00** | 0.98 | **0.69** | **0.73** | 0.25 |
| | ACTIONPIECE-REPLACE | 0.28 | 0.00 | 0.55 | 0.24 | 0.10 | 0.28 |
| MaxEnt-GFlowNet | ACTIONPIECE-INCREMENT | **1.11** | **1.00** | 0.98 | **0.69** | **0.73** | 0.25 |
| | ACTIONPIECE-REPLACE | 0.26 | 0.00 | 0.47 | 0.24 | 0.18 | 0.23 |
| ShortParse-GFlowNet | ACTIONPIECE-INCREMENT | **1.10** | **1.00** | 0.98 | **0.68** | **0.73** | 0.26 |
| | ACTIONPIECE-REPLACE | 0.34 | 0.00 | 0.63 | 0.27 | 0.17 | 0.27 |
| A2C | ACTIONPIECE-INCREMENT | 0.36 | 0.00 | 0.76 | 0.23 | 0.00 | 0.35 |
| | ACTIONPIECE-REPLACE | 0.11 | 0.00 | 0.35 | 0.10 | 0.00 | 0.24 |
| SAC | ACTIONPIECE-INCREMENT | 0.43 | 0.00 | 0.78 | 0.29 | 0.02 | 0.36 |
| | ACTIONPIECE-REPLACE | 0.15 | 0.00 | 0.51 | 0.10 | 0.00 | 0.22 |
| Random Sampler | ACTIONPIECE-INCREMENT | 0.46 | 0.00 | 0.79 | 0.31 | 0.12 | 0.36 |
| | ACTIONPIECE-REPLACE | 0.36 | 0.00 | 0.70 | 0.27 | 0.04 | 0.34 |

The dataset we consider includes all RNA sequences with a reward of 0.85 or higher. From Table 2, it is clear that ACTIONPIECE-INCREMENT consistently results in higher chunk occurrence values for GFlowNet-based samplers (GFlowNet, MaxEnt-GFlowNet, and ShortParse-GFlowNet). In contrast, A2C, SAC, and the random sampler exhibit moderate to low chunk occurrences under ACTIONPIECE-INCREMENT, while ACTIONPIECE-REPLACE yields near-zero median values

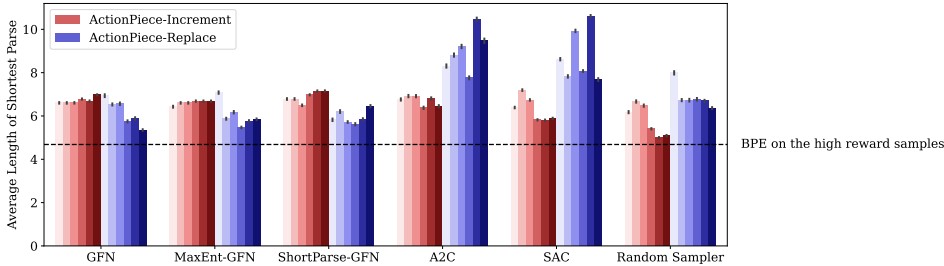

(a) Shortest parse of modes from `L14_RNA1` reward distribution using `L14_RNA1` learned chunks.

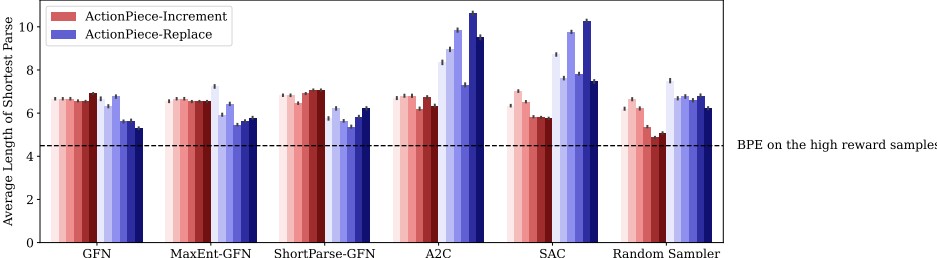

(b) Shortest parse of modes from `L14_RNA2` reward distribution using `L14_RNA1` learned chunks.

Figure 4: **Shortest parse of modes using learned library**. These plots show the average length of the shortest parses for high-reward samples across different models (ShortParse-GFlowNet, MaxEnt-GFlowNet, GFlowNet, SAC, A2C, and Random Sampler) when employing different chunking strategies, across 6 seeds (plotted separately)

.

across all samplers. A similar trend is observed when analyzing chunk coverage, where GFlowNet-based samplers again dominate in terms of learned chunk representation within the high-reward landscape. Moreover, ACTIONPIECE-INCREMENT also outperforms ACTIONPIECE-REPLACE, we speculate because the ACTIONPIECE-REPLACE strategy updates the library too rapidly, making it hard for the policy to model it accurately. These findings suggest that GFlowNet-based samplers augmented with the ACTIONPIECE-INCREMENT effectively capture latent structure of the distribution, particularly around the modes.

**Do chunks reduce description length/complexity of samples?** To evaluate the compressive ability of the chunks, we compute the shortest parse of the (known) modes of the distribution with the learned library of chunks. This is defined as the average length of a trajectory required to sample a mode. Note that the the longest parse of modes would be a trajectory using only atomic actions (letters in string-based environments). We also compute a minimum attainable shortest parse of the modes by performing BPE on the modes directly, which serves as a lower bound.

Figure 4 shows that GFlowNet-based approaches tend to have the lowest short parses of the modes, indicating greater diversity in the learned chunks. In contrast, RL-based methods tend to have higher average parse lengths, as their learned chunks focus on a smaller region of the state space, suggesting that limited exploration can negatively impact the quality of the learned chunks. The chunking mechanism also affects different samplers in opposite ways. For RL-based methods, ACTIONPIECE-REPLACE learns a library that doesn't compress samples very well as evident by the average length of the shortest-parse as opposed to GFN-based methods that do indeed find libraries that capture the latent structure of the distribution and result in a shorter average length compared to ACTIONPIECE-INCREMENT.

**Are chunks transferable to new models and new tasks?** In this section, we study the downstream impact of discovered chunks. We evaluate the downstream performance along two axes: generalization across different but related target distributions, and across different samplers. The aim of this experiment is to identify whether the chunks themselves generalize and which samplers provide the best chunks in terms of downstream performance.

Figure 5 shows that initializing samplers with a learned library of chunks improves the number of modes found compared to not performing any chunking. We see this specifically when samplers use learned library of chunks from `L14_RNA1` to sample from `L14_RNA2` and `L14_RNA3`, showing that the chunks do generalize to structurally similar reward distributions. Zooming in on the performances of each sampler, we see that GFlowNet-induced libraries provide *on-average* better results in terms of mode-discovery than the libraries coming from other samplers. This can be ex-

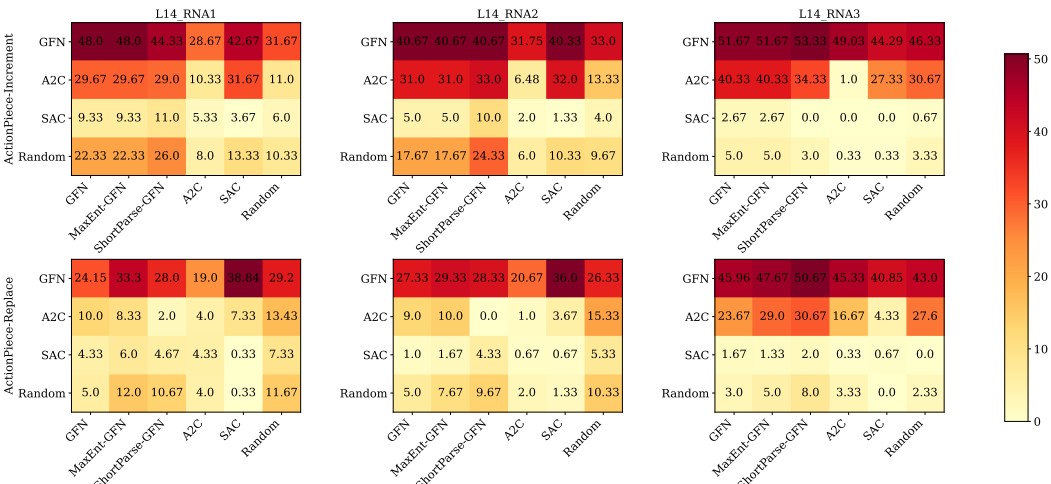

Figure 5: **Donwstream evaluation of discovered chunks**. Each column represents a different environment (L14_RNA1, L14_RNA2, L14_RNA3) and each row presents a chunking mechanism (ACTIONPIECE-INCREMENT and ACTIONPIECE-REPLACE). In each heatmap, we show the number of modes discovered by samplers on the y-axis trained on chunks found by samplers on the x-axis. The color intensity represents the number of modes, with darker shades indicating higher numbers. Average of three random seeds.

plained by the fact that RL-based methods tend to learn chunks specific to a prioritized subspace of the state-space, which fail as a result to generalize to the entire space. While these remarks are true for libraries learned using ACTIONPIECE-INCREMENT, the conclusions do not transfer well to the ACTIONPIECE-REPLACE mechanism which achieves poor performance. We speculate that this mechanism is too abrupt when updating the library, since it updates all chunks at once, whereas ACTIONPIECE-INCREMENT adopts a more curriculum-like strategy of adding only one chunk at a time, allowing the samplers enough time to adapt to the newly added chunk.

## 7 CONCLUSION

Abstraction and dynamic concept learning are critical in human cognition and should be modeled in artificial learning systems. In this paper, we investigated dynamic action space learning in the context of amortized samplers. We proposed ACTIONPIECE, an extensible approach which leverages tokenizers to chunk action sequences, which can be viewed as a strategy for trading reduced depth of the MDP in exchange for increased breadth (Pignatelli et al., 2024). We demonstrated its effectiveness in improving the performance of samplers on a variety of domains with respect to mode discovery and capturing the latent structure of the environment. Our empirical results also indicate that the selected chunks for samplers like GFlowNets capture underlying structure in the target distribution and can generalize to other target distributions. This approach has multiple possible utilities: the learned chunks capture the underlying structure of the reward distribution (and are therefore potentially interpretable), can facilitate transfer to more complex problems where individual trajectories would grow too long if constructed only of atomic actions, and under the right conditions, appears to facilitate exploration itself.

ACTIONPIECE builds on a long line of research into macro-actions (*e.g.*, Durugkar et al. (2016); Iba (1989); Chang et al. (2022); Dulac et al. (2013); Jong et al. (2008)), but previous results have shown they have mixed utility in general. They have not until now been evaluated in the context of diversity-seeking RL approaches such as GFlowNets. Our results suggest that the use of online library construction works best in conjunction with methods that explicitly maximize diversity: they learn libraries that allow for the shorter parse lengths and are more useful for discovering new modes when transferred to new tasks. This points towards an intriguing interaction between the trajectory sampling method and the ability to learn robust, general-purpose macro-actions. Our results suggest that future studies on action abstraction for amortized samplers should explore the interaction between the method for macro-action construction and exploration.

**Limitations and future work.** In this work, we perform chunking with the BPE tokenizer for a fixed target distribution. Future work can study learning task-conditioned libraries, as well as formulating the problem of learning abstractions as modelling the joint distribution $p(\mathcal{X}, \mathcal{L})$, where $\mathcal{L}$ denotes a library of abstractions – that is, modeling a *distribution* over possible concept libraries. Future work can also explore the application of the ideas of abstractions discussed in the paper in the context of code generation, where the right abstractions can naturally make complex tasks easier (Ellis et al., 2021; Stengel-Eskin et al., 2024).

## ACKNOWLEDGEMENTS

The authors acknowledge funding from CIFAR, NSERC, IVADO, NRC and Samsung. The research was enabled by computational resources provided by the Digital Research Alliance of Canada (`https://alliancecan.ca`), Mila (`https://mila.quebec`), and NVIDIA.

## REPRODUCIBILITY STATEMENT

We discuss all the details and hyperparameters used to reproduce the results in the paper in Appendix B. We also include the code to reproduce the experiments along with the submission: https://github.com/GFNOrg/Chunk-GFN.

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

# Appendix

## Table of Contents

## A  ADDITIONAL RELATED WORKS

**Tokenization.**    Our work builds on the principle of tokenization, which is a text pre-processing step for natural language tasks and a critical piece for language models (LMs). A tokenizer produces a vocabulary that can be used by the LLM in order to correctly process text in the language of the training dataset[2]. Byte Pair Encoding (BPE) (Gage, 1994; Sennrich et al., 2015) starts with an initial vocabulary of tokens and successively merges the most frequent pairs of adjacent tokens to get a new composite one. WordPiece (Schuster & Nakajima, 2012) works similarly to BPE where a pair is only merged if the ratio of the likelihood of its joint to that the product of its marginals is the highest. For BPE and WordPiece both, the procedure is repeated until a maximum vocabulary size is reached.

## B  EXPERIMENTAL DETAILS

```python
FractalGrid reward landscape
def create_grid(R0, R1, R2, side_length):
        grid = R0 * np.ones((side_length, side_length))
        depth = int(np.log(side_length) / np.log(2)) - 2

        def fill_r2(x, y, current_size, current_depth):
            if current_depth == 0 or current_size < 1:
                return

            # Ensure we don't go out of bounds
            max_y = min(y, side_length - 1)
            max_x = min(x + current_size - 1, side_length - 1)

            # Fill bottom-left cell
            grid[max_y - 1, x + 1] = R0 + R1 + R2
            grid[max_y - 1, x] = R0 + R1
            grid[max_y, x + 1] = R0 + R1
            grid[max_y, x] = R0 + R1

            if current_depth < depth:
                # Fill bottom-right cell
                grid[max_y - 1, max_x - 1] = R0 + R1 + R2
                grid[max_y - 1, max_x] = R0 + R1
                grid[max_y, max_x - 1] = R0 + R1
                grid[max_y, max_x] = R0 + R1

                # Fill top-left cell
                grid[y - current_size + 2, x + 1] = R0 + R1 + R2
                grid[y - current_size + 2, x] = R0 + R1
                grid[y - current_size + 1, x + 1] = R0 + R1
                grid[y - current_size + 1, x] = R0 + R1

                # Calculate the size and position of the next level
                next_size = current_size // 2
                next_x = x + next_size
                next_y = y - next_size

                # Recursive call for the next level
                fill_r2(next_x, next_y, next_size, current_depth - 1)

        # Start from the bottom-left corner
        fill_r2(0, side_length - 1, side_length, depth)
        grid = torch.from_numpy(grid[::-1].copy()).float()
        return grid
```

Listing 1: Code for implementing the FractalGrid reward landscape.

---

[2]Token-free approaches (Xue et al., 2021) do exist and only make use of bytes/characters.

In this section, we provide implementation details along with necessary the hyperparameters for reproducing the experiments presented in the paper. All runs use a single GPU with runs taking up to 36 hours maximum.

## B.1 FRACTALGRID

Our implementation is inspired by hypergrid that was introduced in Bengio et al. (2021). In this study, we consider only the 2D case and present results for three settings with increasing grid size: $65^2$, $129^2$ and $257^2$. The main difference lies in the reward landscape which we have altered. We show the code used to compute the reward landscape (see Code 1).

**Environment.** The starting state is $s_0 = (0, 0) \in \mathbb{R}^2$ and the atomic action space consists of the following actions: UP, RIGHT and <EXIT>. The sampler can choose to exit the trajectory at any stage. We chose $R_0 = 0.1$, $R_1 = 0.5$ and $R_2 = 2$. We run the samplers for a total of 31250 iterations and a batch size of 64, adding up to a total of 2 million visited states during training.

**Architecture.** The forward policy in SAC, A2C, Option-Critic and GFlowNet is implemented using a feedforward network with 3 layers and a hidden dimension of 128. The last layer produces an action embedding of dimension 128 as well and a layer norm is applied just before (Xu et al., 2019). For the activation function, we use ReLU (Nair & Hinton, 2010). For A2C, Option-Critic and GFlowNet, the forward policy has a learning rate of $10^{-4}$ whereas for SAC, it is $3 \times 10^{-4}$. For GFlowNet, we use an initial value for the learnable log-partition value of 90 and a learning rate of $10^{-3}$. The critic in A2C and Option-Critic is also parametrized by a feedforward network with 3 layers and a hidden dimension of 128 where a layer norm is applied before the last layer. The critic in A2C and Option-Critic has a learning rate of $10^{-4}$ whereas in SAC, it has a learning rate of $3 \times 10^{-4}$. The termination probability network is parametrized using a feedforward network with 3 layers and a hidden dimension of 128 with a learning rate of $10^{-4}$.

**Samplers.** Both GFlowNet and SAC are off-policy algorithms. For both, we use a prioritized replay buffer with diversity constraints (see section on replay buffer). Our diversity criterion is the hamming distance between final states and the threshold is set to 1. The replay buffer capacity is set to 1000 due to the small size of the state space. During training, 55% of samples come from the replay buffer and the rest from the sampler. We don't use $\epsilon$-greedy exploration for SAC, but for GFlowNet we linearly decay $\epsilon$ from 0.5 down to 0.1. SAC uses an entropy coefficient of 0.2 and A2C and Option-Critic use an entropy coefficient of 0.5. For GFlowNet, we use a reward exponent of 1. We use 10 options for the Option-Critic and a coefficient of 0.01 for regularizing the termination probability.

**Chunking.** We perform chunking every 1250 iterations to get 25 chunks at the end of training.

## B.2 RNA BINDING

We consider sampling sequences of length 14 and sequences of length 50. The reward is scaled between $10^{-10}$ and 1.

**Environment** The starting state is the empty string and the atomic action space consists of the following actions: A, C, G, U which append the nucleobases to the current string and <EOS> for End-Of-String. We run the samplers for a total of 25000 iterations and a batch size of 64, adding up to a total of 1.6 million visited states during training.

**Architecture** The forward policy in SAC, A2C, Option-Critic and GFlowNet is implemented using an LSTM (Hochreiter & Schmidhuber, 1997) with 2 layers and a hidden dimension of 128 followed by a feedforward network with 2 layers. The last layer produces an action embedding of dimension 128 as and a layer norm is applied just before (Xu et al., 2019). For the activation function, we use ReLU (Nair & Hinton, 2010). For A2C, Option-Critic and GFlowNet, the forward policy has a learning rate of $10^{-4}$ whereas for SAC, it's $3 \times 10^{-4}$. For GFlowNet, we use an initial value for the learnable log-partition value of 11 for length 14 and 22 for length 50, the learning rate is set to $10^{-3}$. The critic in A2C and Option-Critic is also parametrized by a two layer LSTM followed by a two layer feedforward network of hidden dimension of 128 where a layer norm is applied before the last layer. The critic in A2C and Option-Critic has a learning rate of $10^{-4}$ whereas in SAC, it has a learning rate of $3 \times 10^{-4}$. The termination probability network is parametrized LSTM (Hochreiter & Schmidhuber, 1997) with 2 layers and a hidden dimension of 128 followed by a feedforward network with 2 layers and a learning rate of $10^{-4}$.

**Samplers.** Both GFlowNet and SAC are off-policy algorithms. For both, we use a prioritized replay buffer with diversity constraints (see section on replay buffer). Our diversity criterion is the

hamming distance between final states and the threshold is set to 3 for length 14 and 10 for length 50. The replay buffer capacity is set to 10000. During training, 55% of samples come from the replay buffer and the rest from the sampler. We don't use $\epsilon$-greedy exploration for SAC, but for GFlowNet we linearly decay $\epsilon$ from 0.1 down to 0.01. SAC uses an entropy coefficient of 0.1 for length 14 and 0.025 for length 50. A2C and Option-Critic use an entropy coefficient of 0.05 for length 14 and 0.005 for length 50. For GFlowNet, we use a reward exponent of 10 for length 14 and 75 for length 50. We use 10 options for the Option-Critic and a coefficient of 0.01 for regularizing the termination probability.

**Chunking.** We perform chunking every 1000 iterations which means that at the end of the training, we have 25 chunks. For GFlowNet-based samplers we use a different approach where we apply chunking once the TB loss gets below certain threshold. For length 14 the initial loss threshold is set to 1 whereas for length 50 it is set to 5. Once chunking is applied, the loss threshold is multiplied by 0.75.

### B.3 BIT SEQUENCE

We modify the original bit-sequence task introduced in Malkin et al. (2022) to have a different reward function. We first initialize a list of "words": [00000000, 11111111, 11110000, 00001111, 00111100]. The reward function is then designed to favor strings that contain a maximum number of these words (including repeated ones). As such, the reward function is:

$$R(s) = \frac{\text{number of words in } s}{N/8} \tag{5}$$

Where $N$ is the sequence length and 8 is the size of the words. $N/8$ represents the maximum number of words in a string of length $N$. The numerator is computed using dynamic programming. We consider two settings: Sampling sequences of length 64 and those of length 128.

**Environment.** The starting state is the empty string and the atomic action space consists of the following actions: 0, 1 and <EOS> for End-Of-String. We run the samplers for a total of 25000 iterations and a batch size of 64, adding up to a total of 1.6 million visited states during training.

**Architecture.** The forward policy in SAC, A2C, Option-Critic and GFlowNet is implemented using an LSTM (Hochreiter & Schmidhuber, 1997) with 2 layers and a hidden dimension of 128 followed by a feedforward network with 2 layers. The last layer produces an action embedding of dimension 128 as and a layer norm is applied just before (Xu et al., 2019). For the activation function, we use ReLU (Nair & Hinton, 2010). For A2C, Option-Critic and GFlowNet, the forward policy has a learning rate of $10^{-4}$ whereas for SAC, it is $3 \times 10^{-4}$. For GFlowNet, we use an initial value for the learnable log-partition value of 40 for length 64 and 30 for length 128, the learning rate is set to $10^{-3}$. The critic in A2C is also parametrized by a two layer LSTM followed by a two layer feedforward network of hidden dimension of 128 where a layer norm is applied before the last layer. The critic in A2C has a learning rate of $10^{-4}$ whereas in SAC, it has a learning rate of $3 \times 10^{-4}$. The termination probability network is parametrized LSTM (Hochreiter & Schmidhuber, 1997) with 2 layers and a hidden dimension of 128 followed by a feedforward network with 2 layers and a learning rate of $10^{-4}$.

**Samplers.** Both GFlowNet and SAC are off-policy algorithms. For both, we use a prioritized replay buffer with diversity constraints (see section on replay buffer). Our diversity criterion is the hamming distance between final states and the threshold is set to 8 for length 64 and 16 for length 128. The replay buffer capacity is set to 10000. During training, 55% of samples come from the replay buffer and the rest from the sampler. We don't use $\epsilon$-greedy exploration for SAC, but for GFlowNet we linearly decay $\epsilon$ from 0.1 down to 0.01. SAC uses an entropy coefficient of 0.01 for length 64 and 0.005 for length 128. A2C and Option-Critic use an entropy coefficient of 0.05 for length 64 and 0.01 for length 128. For GFlowNet, we use a reward exponent of 100 for length 64 and 200 for length 128. We use 10 options for the Option-Critic and a coefficient of 0.01 for regularizing the termination probability.

**Chunking** We perform chunking every 1000 iterations to get 25 chunks at the end of training.

### B.4 GRAPH

In this paper, we departed from the usual way graph environments are set-up. Indeed, for chunking to make sense and essentially work for a graph environment, we need to be careful about the design of the MDP. For a modular structure to emerge in the action sequence, we need the action space to reflect that. In previous work, actions for adding edges would be to select any two nodes and connect them. However, connecting node 3 and 4 although similar in structure to connecting node 5

and 6, would be seen as different for the tokenizer. Let $N$ denote the maximum allowed number of nodes in the graph, we build the action space and its constraints as follows:

1. For an empty graph:
   - Only the `ADD-NODE` action is permitted.
2. For a non-empty graph:
   (a) If the graph contains exactly one node:
      - Choose between `<EOG>` (end of graph generation) or `ADD-NODE`.
   (b) If the graph contains two or more nodes:
      i. When the last added node is connected:
         - Choose from `<EOG>`, `ADD-NODE`, or connect the last node to a previous node.
         - Edge addition actions are formatted as `ADD-EDGE-i`, where $i \in \{-1, -2, ..., -(k - j + 1)\}$.
         - Here, $k$ is the order of the last added node, and $j$ is the order of the farthest connected node to the last one.
      ii. When the last added node is not connected:
         - You must connect it to one of the previous nodes.
         - Allowed actions are `ADD-EDGE-i`, where $i \in \{-1, -2, ..., -k\}$ where $k$ is the order of the last node.

This structure ensures that edge additions are always relative to the last added node, creating a consistent and potentially interpretable action space. For instance, `ADD-EDGE-(-1)` always means connecting the last node to the previous one, regardless of their absolute node numbers. We speculate that an objective of future work could be to explicitly produce chunks with interpretable properties, for example, using predefined rules which favors the retention of particular chunks, or using scores from human feedback. By defining actions in this relative manner, we create an action space that is more amenable to finding recurring structure in it. The reward function is defined as the total number of cycles divided by the maximum number of cycles that can be found in a graph with $N$ nodes where $N$ is the maximum number of nodes in the graph:

$$R(s) = \frac{\text{number of cycles in graph } s}{\frac{N(N-1)}{2} - N + 1} \tag{6}$$

**Environment.** The starting state is the empty graph and the atomic action space consists of the following actions: `ADD-NODE`, `ADD-EDGE-k` where an edge is added between the last node and the k-th node before it. k takes values in the set $\{-1, -2, \ldots, -N + 1\}$ where $N$ is the maximum number of nodes. The last action is `<EOG>` for End-Of-Graph. We run the samplers for a total of 25000 iterations and a batch size of 64, adding up to a total of 1.6 million visited states during training.

**Architecture.** The forward policy in SAC, A2C and GFlowNet has two Graph Attention Layers (Veličković et al., 2017; Brody et al., 2021) and a hidden dimension of 128 followed by a feedforward network with 2 layers. The last layer produces an action embedding of dimension 128 as and a layer norm is applied just before (Xu et al., 2019). For the activation function, we use ReLU (Nair & Hinton, 2010). For A2C and GFlowNet, the forward policy has a learning rate of $10^{-4}$ whereas for SAC, it is $3 \times 10^{-4}$. For GFlowNet, we use an initial value for the learnable log-partition value of 10 for $N = 7$ and 30 for length $N = 10$, the learning rate is set to $10^{-3}$. The critic in A2C is also parametrized by a two layer LSTM followed by a two layer feedforward network of hidden dimension of 128 where a layer norm is applied before the last layer. The critic in A2C has a learning rate of $10^{-4}$ whereas in SAC, it has a learning rate of $3 \times 10^{-4}$.

**Samplers.** Both GFlowNet and SAC are off-policy algorithms. For both, we use a prioritized replay buffer with diversity constraints (see section on replay buffer). Our diversity criterion is the hamming distance between the graphs adjacency matrices and the threshold is set to 3 for $N = 7$ and 5 for $N = 10$. The replay buffer capacity is set to 10000. During training, 55% of samples come from the replay buffer and the rest from the sampler. We don't use $\epsilon$-greedy exploration for SAC, but for GFlowNet we linearly decay $\epsilon$ from 0.1 down to 0.01. SAC uses an entropy coefficient of 0.1. A2C uses an entropy coefficient of 0.05. For GFlowNet, we use a reward exponent of 1.

**Chunking.** We perform chunking every 1000 iterations to get 25 chunks at the end of training.

## B.5 REPLAY BUFFER

In this section, we provide additional details about the replay buffer used throughout the paper (see Algorithm 2). Our replay buffer maintains a balance between having high-reward samples and

diverse ones. When a new batch of elements is added, we only keep the ones that have a reward higher than the minimal reward in the buffer. This ensures that the minimum reward in the buffer never decreases, meaning that we don't hinder the quality of the samples in the buffer.

Once the high-reward states of the batch are kept, we iterate through each state $s$:

- Let $d(s, B) = \min_{s' \in B} d(s, s')$ where $d$ is a distance metric. Given a cutoff distance $d_c$ that controls how much diversity we want, we only add $s$ to the buffer $B$ if $d(s, B) > d_c$.
- If the state $s$ is not diverse enough, we look for the most similar state $s_b = \arg\min_{s' \in B} d(s, s')$ to it in the buffer and if $s$ state has a higher reward than $s_b$, we replace $s_b$ by $s$.

Finally, we sort the buffer by the reward and truncate it to capacity.

---

**Algorithm 2** Adding trajectories to the Prioritized Replay Buffer

---

**Require:** New experiences $E$, buffer $B$, cutoff distance $d_c$
1: **if** $|B| <$ capacity **then**
2:     Add $E$ to $B$
3:     Sort $B$ by log-reward
4: **else**
5:     $E' \leftarrow e \in E : r_e \geq \min(r_b), b \in B$
6:     **for** each $e \in E'$ **do**
7:         **if** $\min(\text{distance}(e, b)) > d_c, \forall b \in B$ **then**
8:             Add $e$ to $B$
9:         **else if** $r_e > r_b$ for $b = \arg\min(\text{distance}(e, b))$ **then**
10:            Replace $b$ with $e$ in $B$
11:         **end if**
12:     **end for**
13:     Sort $B$ by log-reward
14:     Truncate $B$ to capacity
15: **end if**

---

### B.6 SHORTPARSE

In this section, we describe in detail how ShortParse $P_B$ is computed. We only use ShortParse for string-based environment for its tractability, so the focus will be on DAGs that generate strings. Let's introduce some notation first:

> **Notation:**
> - $A$: the alphabet of characters.
> - $V$: the set of tokens. We will not assume $A \subseteq V$ unless stated, in which case the inclusion of $A$ in $V$ forms the set of "atomic tokens".
> - $\mathcal{X}$: the set of terminal sequences.
> - $\mathcal{S} \supseteq \mathcal{X}$: state space, the set of sequences that can be reached on the way to generating a sequence in $\mathcal{X}$.
> - $\mathcal{T}$: the set of trajectories (sequences of tokens that when concatenated give some $x \in \mathcal{X}$).
> - $\mathcal{T}_s$ ($s \in \mathcal{S}$): the set of sequences of tokens that when concatenated give. Note $\mathcal{T} = \bigcup_{x \in \mathcal{X}} \mathcal{T}_x$.
> - $|s|$ ($s \in \mathcal{S}$ or $s \in V$): the length of $s$.
> - $|\tau|$ ($\tau \in \mathcal{T}$): the number of tokens in trajectory $\tau$. Note that in general for $\tau \in \mathcal{T}_s$, $|\tau| \leq |s|$.
> - $s{:}i$ ($s \in \mathcal{S}, 0 \leq i \leq |s|$): the initial substring of $s$ of length $i$.

Let $N(s) := |\mathcal{T}_s|$. For a given $s$, we can compute $N(s{:}i)$ for $i = 0, \ldots, |s|$ by the following recurrence:

$$N(s{:}0) = 1,$$

$$N(s{:}i) = \sum_{t \in V : s{:}i \text{ ends in } t} N(s{:}i-|t|) \qquad (i > 0).$$

This computation is linear in the sequence length and can be done incrementally as $s$ is being constructed by apposition of tokens. That is, one maintains along with $s$ an array containing $N(s{:}i)$ for $0 \leq i \leq |s|$. When a token $t$ is appended to $s$, we append to this array the newly computed $N(s{:}i+j)$ for $j = 1, \ldots, |t|$.

It is also possible to compute a weighted version. Let

$$N_\lambda(s) = \sum_{\tau \in \mathcal{T}_s} e^{\lambda|\tau|},$$

so $N(s) = N_0(s)$. This can be computed by the recurrence:

$$N_\lambda(s_{:0}) = 1,$$

$$N_\lambda(s_{:i}) = e^\lambda \sum_{t \,\in\, V:\, s_{:i} \text{ ends in } t} N(s_{:i-|t|}) \qquad (i > 0).$$

**A family of length-dependent Markovian backward policies**  For any $\lambda \in \mathbb{R}$, we define a backward policy $P_B^\lambda$, as a conditional distribution over trajectories, by

$$P_B^\lambda(\tau \mid x) = \frac{e^{\lambda|\tau|}}{N_\lambda(x)} \propto e^{\lambda|\tau|} \quad (x \in \mathcal{X}, \tau \in \mathcal{T}_§).$$

**Proposition:** This policy is Markovian, and its stepwise factorization is given by

$$P_B^\lambda(s \mid st) = \frac{e^\lambda N_\lambda(s)}{N_\lambda(st)} \quad (s \in \mathcal{S}, t \in V). \tag{7}$$

**Proof:** First, we have to check that this stepwise policy is well-defined, that is, sums to 1 over parents of any state $st$. This can be seen from the recurrence for $N_\lambda$ above.

Now, suppose $x$ is the concatenation of tokens in a trajectory $\tau = t_1 t_2 \ldots t_n$. Then the expression for $P_B(\tau \mid x)$ given by the product of stepwise transitions is

$$\frac{e^\lambda N_\lambda(\emptyset)}{N_\lambda(t_1)} \frac{e^\lambda N_\lambda(t_1)}{N_\lambda(t_1 t_2)} \cdots \frac{e^\lambda N_\lambda(t_1 \ldots t_{n-1})}{N_\lambda(t_1 \ldots t_n)} = \frac{e^{n\lambda}}{N_\lambda(x)},$$

which exactly matches the trajectory-wise definition of $P_B^\lambda$ above. $\square$

We note the following:

- The stepwise factorization of $P_B^\lambda$ can be useful, for example, if using a forward-looking objective.
- As a special case, with $\lambda = 0$ we have the maximum-entropy backward policy as studied in Mohammadpour et al. (2024), which is uniform over trajectories conditioned on the terminal state – notably different from the backward policy that is uniform at each transition. In that paper, the numbers of trajectories to every state had to be learned, but in our setting, computation is efficient (see above) and no learning is necessary.
- For $\lambda < 0$, shorter trajectories are preferred, and as $\lambda \to -\infty$, we approach a policy that is peaky on optimal tokenizations. Conversely, if $\lambda \to +\infty$ and $A \subseteq V$, we approach a policy that only tokenizes into atomic tokens.

For ShortParse, we use $\lambda = -5$ in all of the paper.

### B.7 Sampling quality metrics

In subsection 6.1, we computed the quality of the learned distribution using three metrics. We detail them here:

**JSD.**  This is the Jensen–Shannon divergence, it measures the similarity between two distributions and is symmetric and takes values in $[0, 1]$. Given two distributions $p$ and $q$, it is defined as follows:

$$JSD(p||q) = \frac{1}{2} D_{KL}(p||M) + \frac{1}{2} D_{KL}(q||M) \tag{8}$$

Where $D_{KL}$ is the Kullback-Leibler divergence and $M = \frac{1}{2}(p + q)$.

**L1 distance.**  This computes the L1 norm between two distributions $p$ and $q$ and is double the total variation distance:

$$||p - q||_1 = \sum_x |p(x) - q(x)| \tag{9}$$

Table 3: Number of modes discovered during training in the `L14_RNA1` environment. The ATOMIC case serves a baseline for comparing BPE, WordPiece and Uniform tokenizers for both mechanisms. Greener shades indicate greater improvement over the ATOMIC baseline, while redder shades represent a decline in performance.

| Sampler | ATOMIC | ACTIONPIECE-INCREMENT | | | ACTIONPIECE-REPLACE | | |
|---|---|---|---|---|---|---|---|
| | — | BPE | Uniform | WordPiece | BPE | Uniform | WordPiece |
| GFlowNet | $47.12_{\pm4.18}$ | $49.00_{\pm1.15}$ | $46.33_{\pm2.49}$ | $50.00_{\pm1.41}$ | $45.54_{\pm5.09}$ | $42.33_{\pm2.87}$ | $49.67_{\pm0.47}$ |
| MaxEnt-GFlowNet | $47.12_{\pm4.18}$ | $48.83_{\pm0.69}$ | $37.33_{\pm7.54}$ | $49.67_{\pm0.47}$ | $45.00_{\pm3.37}$ | $27.33_{\pm3.68}$ | $48.33_{\pm1.70}$ |
| ShortParse-GFlowNet | $47.12_{\pm4.18}$ | $48.97_{\pm1.34}$ | $39.00_{\pm7.12}$ | $49.67_{\pm0.94}$ | $43.01_{\pm6.89}$ | $22.00_{\pm8.60}$ | $50.33_{\pm0.47}$ |
| A2C | $36.67_{\pm180}$ | $23.50_{\pm0.96}$ | $23.67_{\pm4.78}$ | $26.67_{\pm0.47}$ | $18.00_{\pm3.51}$ | $25.67_{\pm0.94}$ | $19.67_{\pm2.49}$ |
| SAC | $19.50_{\pm3.49}$ | $1.17_{\pm1.34}$ | $0.00_{\pm0.00}$ | $1.67_{\pm1.25}$ | $3.50_{\pm2.22}$ | $4.33_{\pm2.05}$ | $1.33_{\pm1.25}$ |
| Option-Critic | $16.83_{\pm1.95}$ | — | — | — | — | — | — |
| Random Sampler | $0.17_{\pm0.37}$ | $18.67_{\pm5.25}$ | $1.00_{\pm1.41}$ | $28.67_{\pm1.25}$ | $18.17_{\pm3.24}$ | $0.33_{\pm0.47}$ | $19.67_{\pm1.70}$ |

**ELBO Gap.** Following past work (Lahlou et al., 2023; Zhang & Chen, 2022; Sendera et al., 2024), we compute a variational lower bound on the log-partition function $\log Z = \sum_x R(x)$:

$$\log Z = \sum_x R(x)$$

$$= \log \mathbb{E}_{\tau=(\ldots\to x)\sim P_F} \left[ \frac{R(x)P_B(\tau|x)}{P_F(\tau)} \right]$$

$$\geq \mathbb{E}_{\tau=(\ldots\to x)\sim P_F} \log \left[ \frac{R(x)P_B(\tau|x)}{P_F(\tau)} \right]$$

Where $P_B$ and $P_F$ are the backward and forward policy respectively. Thus $\log \hat{Z} = \frac{1}{K}\sum_{\tau=(\ldots\to x)\sim P_F} \log\left[\frac{R(x)P_B(\tau|x)}{P_F(\tau)}\right]$ that represents the lower bound estimate, is computed using $K = 10000$ trajectories in this paper for all tasks. The ELBO gap is hence defined as:

$$\text{ELBO Gap} \triangleq \left| \log Z - \log \hat{Z} \right|$$

$$= \left| \log Z - \frac{1}{K} \sum_{\tau=(\ldots\to x)\sim P_F} \log \left[ \frac{R(x)P_B(\tau \mid x)}{P_F(\tau)} \right] \right|$$

Note that we only compute the above quantity for environments where the ground-truth partition function is tractable to compute.

# C   LEARNED LIBRARIES OF CHUNKS

In this section, we show the chunks learned by each sampler for both chunking mechanisms.

## C.1   FRACTALGRID

Figure 7 shows a subset of chunks found using the ACTIONPIECE-INCREMENT mechanism. All samplers effectively learn diverse "paths" and these range from straight lines to a *zigzag*-like shape. These *zigzag* shapes allow the sampler to get from the initial state to the first mode in the middle of the grid. Note that straight UP and RIGHT chunks can also lead to the modes by composing them together. This presents a straight-forward solution where the sampler can choose to first get the $x$ coordinates right by just using the straight line chunks towards the right and then switch to using straight line chunks going up.

## C.2   RNA BINDING

Figure 8 shows the frequency of usage of chunks for different samplers for both chunking mechanisms. A big difference between both chunking mechanisms that seems to be true for all samplers, is that the size of learned chunks is bigger which makes the learning not much more efficient than using the original ATOMIC baseline. Indeed in Figure 8b, a lot of these chunks are rarely used compared to ACTIONPIECE-INCREMENT. When it comes to samplers however, regardless of the chunking mechanism, GFN-related approaches seem to learn shorter chunks on average compared to the other samplers which in turn compresses the trajectories enough to cover the whole state-space as opposed to RL-based methods.

## C.3   BIT SEQUENCE

Figure 9 shows the frequency of use of learned chunks for the bit sequence task for GFN-related approaches using the ACTIONPIECE-INCREMENT mechanism. We can see that most of the library comprises of short chunks with the exception of GFN and ShortParse-GFN that seem to have "mem-

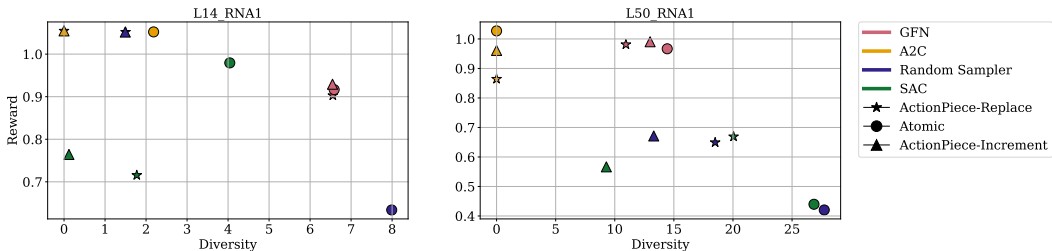

(a) Average reward and diversity for the top 100 samples for the RNA Binding environment.

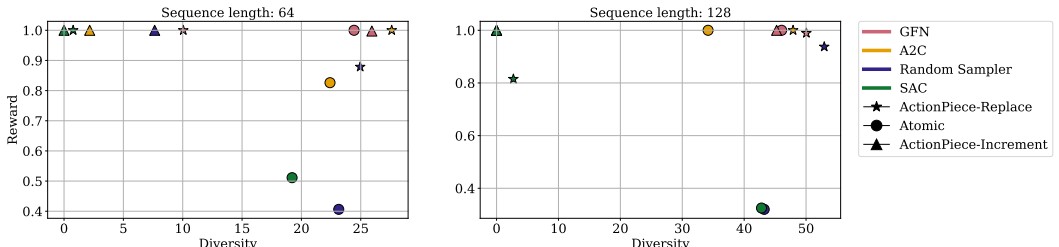

(b) Average reward and diversity for the top 100 samples for the Bit Sequence environment.

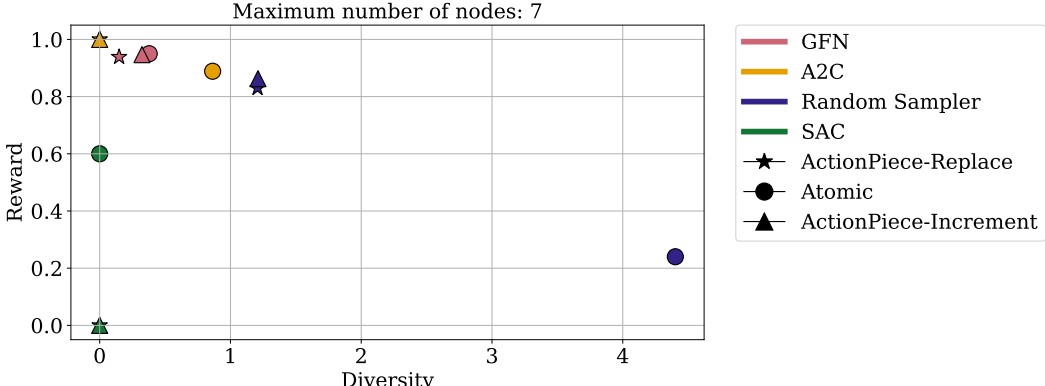

(c) Average reward and diversity for the top 100 samples for the Graph environment.

Figure 6: **Quality of high-reward samples.** Average reward and diversity for the top 100 samples from 10000 generated objects from all samplers and all chunking mechanisms in the RNA Binding, Bit Sequence and Graph environments.

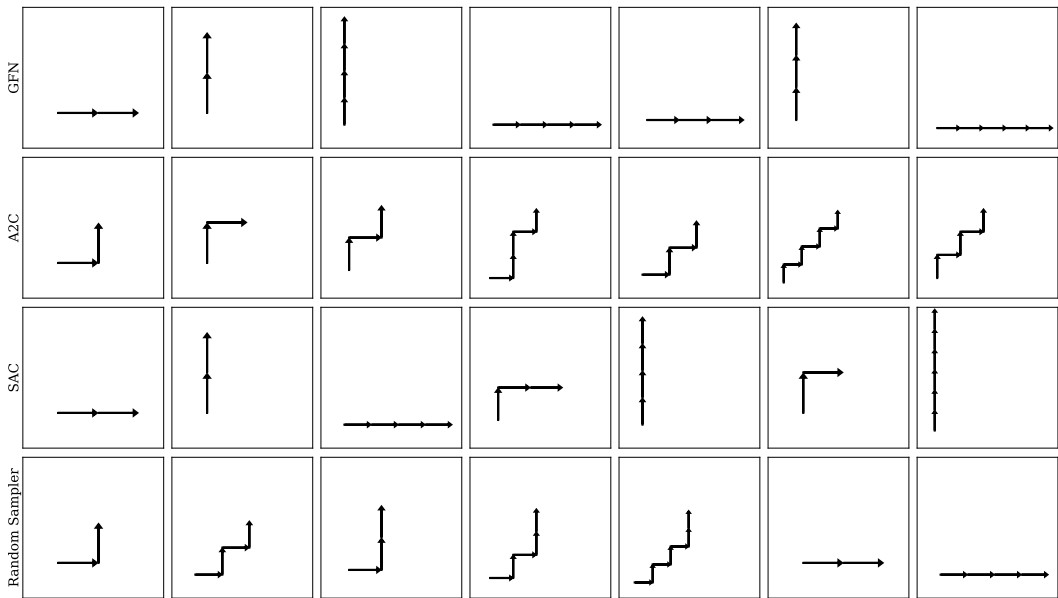

Figure 7: **Chunks visualization.** Visualization of a subset of chunks learned by the samplers for the FractalGrid environment of size $65^2$ with the ACTIONPIECE-INCREMENT mechanism.

orized" one of the modes as part of their learned library. For A2C and SAC however, they seem to have learned chunks that generate a single mode that contains only 1. This can be seen in Figure 10 where A2C doesn't use the action 0 at all and SAC sampling mostly a big part of the same mode. The Random Sampler given as a reference keeps a balance in exploration although it also seems to have added one of the modes to its library. For the ACTIONPIECE-REPLACE mechanism, the trend seems to be reversed, where GFN approaches discover shorter chunks than they had under the ACTIONPIECE-INCREMENT mechanism (see Figure 11) and A2C being much more diverse under ACTIONPIECE-REPLACE. Although SAC learned a diverse library of chunks, it stuck to one mode for its last round of chunking as demonstrated in Figure 12.

### C.4 GRAPH

In this section, we won't show the frequency of all used graph chunks since it is harder to visualize. Instead, we will show some examples of learned subgraphs. We can see from Figure 13 that all samplers are able to learn structures that maximize the number of cycles in the generated graph, which is the reward function we used. All samplers effectively learn meaningful fragments. However, we can see that A2C already "overfits" the reward by seeking chunks that maximize the reward without regards to the diversity whereas this is not the case for SAC, Random Sampler and GFN.

### D THE ROLE OF TOKENIZATION STRATEGY

In this section, we ablate the role of the tokenization strategy by comparing Byte Pair Encoding (BPE), WordPiece, and a *uniform tokenizer* as a baseline. The uniform tokenizer does not require a dataset; it builds new chunks by randomly sampling two tokens from the library and concatenating them. As shown in Table 3, the uniform tokenizer performs worse than both BPE and WordPiece in terms of the number of modes discovered during training for the L14_RNA1 task. Its performance deteriorates further with the ACTIONPIECE-REPLACE mechanism compared to ACTIONPIECE-INCREMENT.

On the other hand, WordPiece outperforms BPE, showing particularly strong performance with the ACTIONPIECE-REPLACE mechanism. We conclude that the choice of tokenization algorithm is crucial, and extra care should be taken when selecting an appropriate tokenizer.

### E SAMPLER EFFECTIVENESS IN SAMPLING HIGH-REWARD OBJECTS

In this section, we look at the top high-reward samples for all samplers as well as their diversity. Starting with RNA Binding, we can see from Figure 6a, that RL-based methods see their average reward of their distribution tail increase with chunking while the diversity of the samples decreasing.

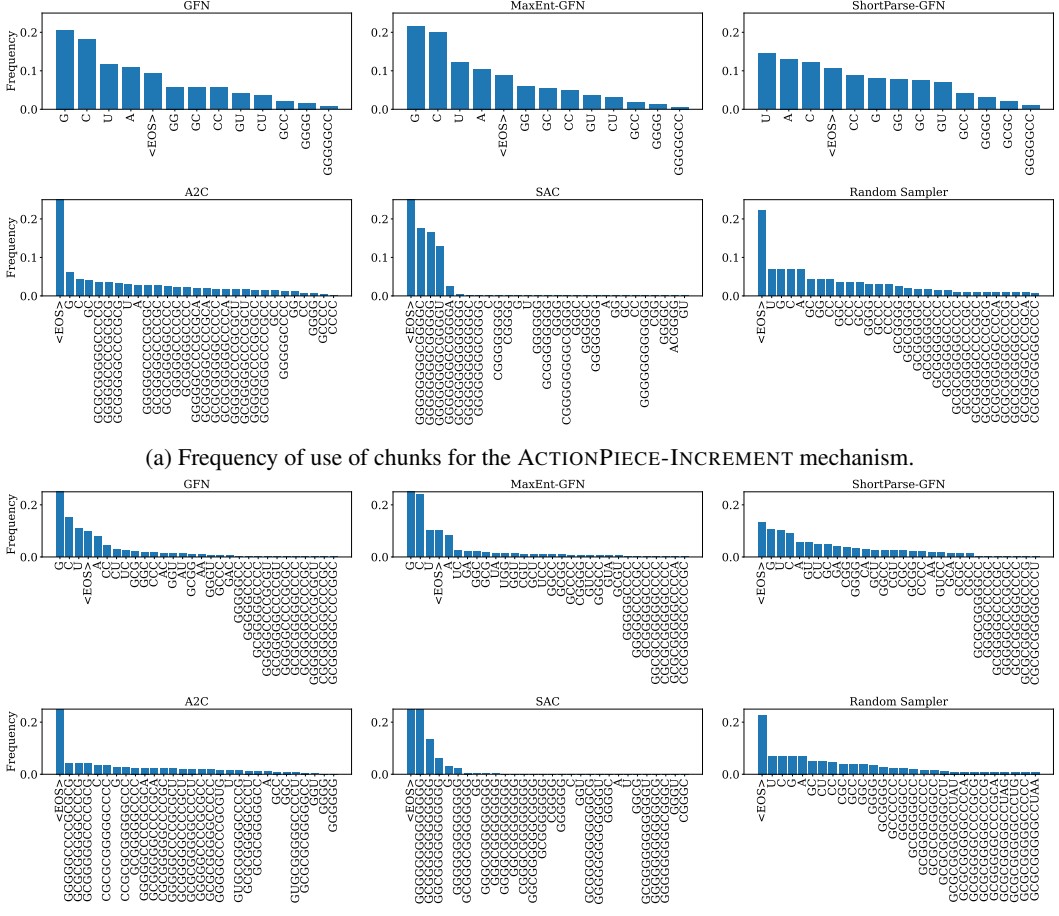

(a) Frequency of use of chunks for the ACTIONPIECE-INCREMENT mechanism.

(b) Frequency of use of chunks for the ACTIONPIECE-REPLACE mechanism.

Figure 8: **Frequency of use of chunks.** Frequency of usage of chunks in the `L14_RNA1` environment for both chunking mechanisms.

This highlights a major flaw of chunking on RL-based methods: They create a negative feedback-loop that harms diversity, which in turn gives chunks concentrated around a small region of the search space which itself further harms exploration. While this is true for RL methods, it is clearly not the case for GFlowNet since we can see from the figure that the diversity of the top-100 samples doesn't decrease with chunking. For bit sequence, the conclusion is similar. Indeed, one can see from Figure 6b that chunking harms RL-methods diversity and increases the average reward. For GFlowNet, the diversity actually increased with chunking. This may be attributed to the structure of the reward distribution where modes (corresponding to a reward of 1) are themselves diverse. For the Graph environment (see Figure 6c), the previous observations stands as well, although for GFlowNet, it seems that diversity is hurt as well in this case.

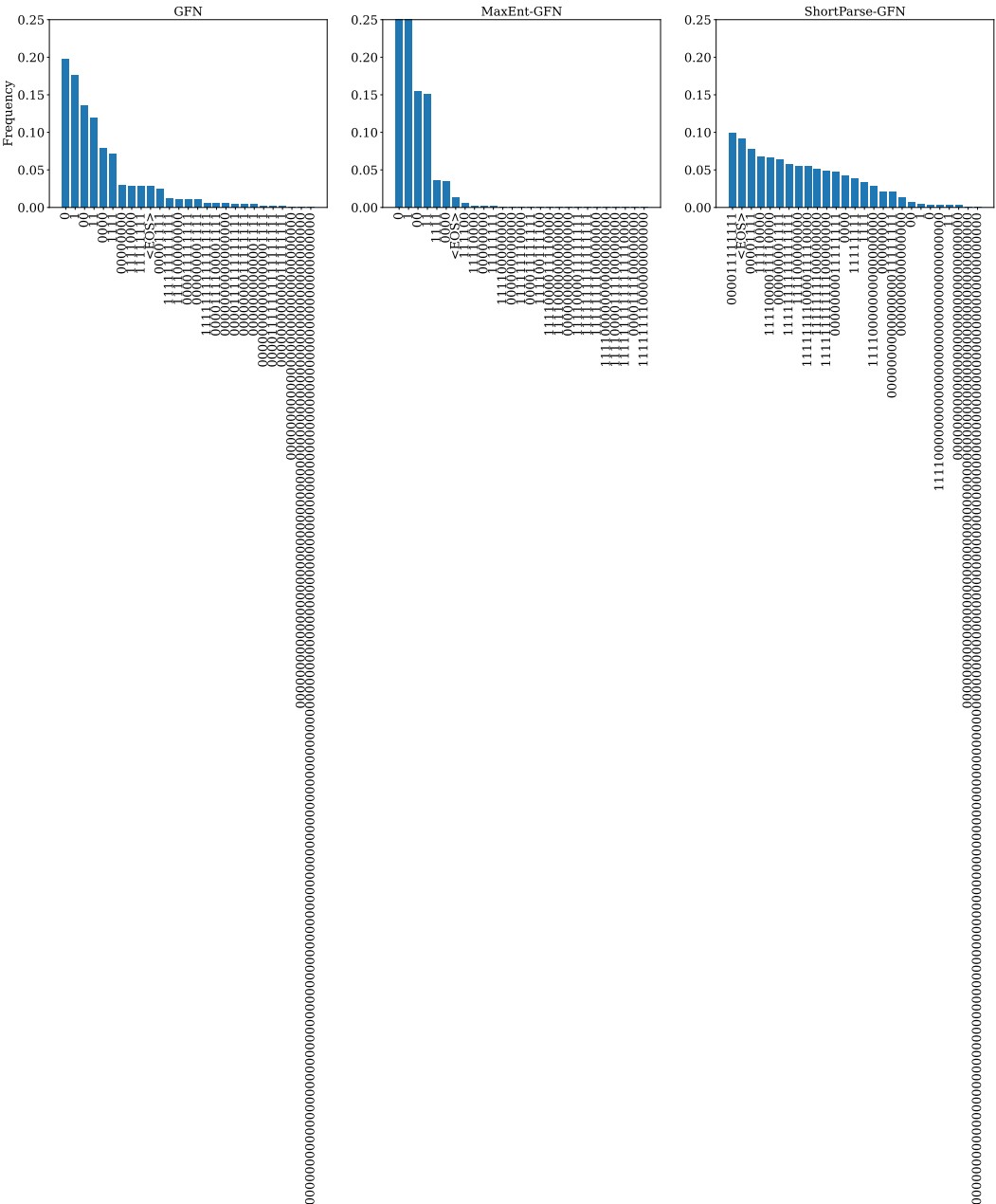

Figure 9: Frequency of use of chunks for the ACTIONPIECE-INCREMENT mechanism for GFlowNet in bit sequence for length 128.

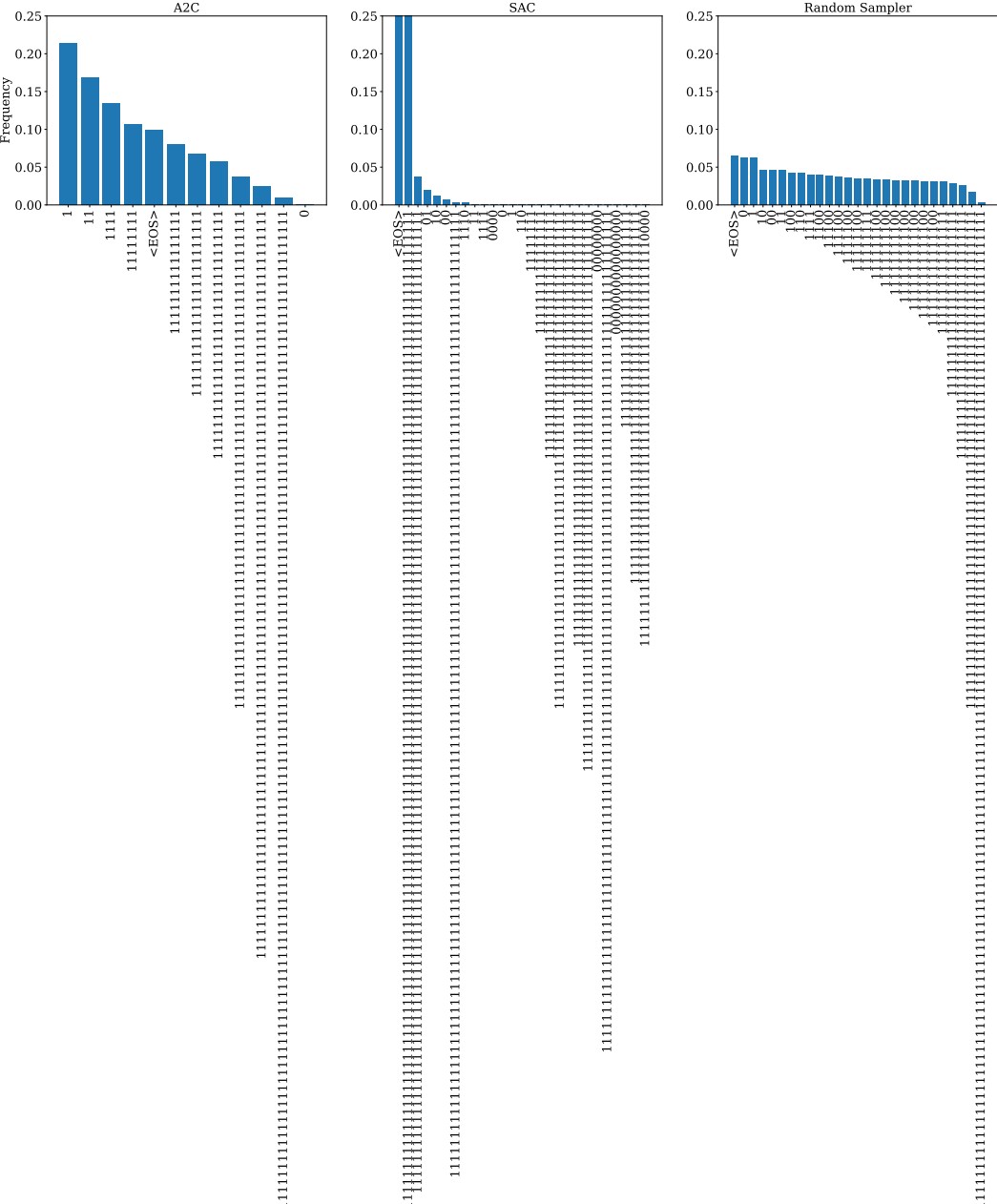

Figure 10: Frequency of use of chunks for the ACTIONPIECE-INCREMENT mechanism for RL methods in bit sequence for length 128.

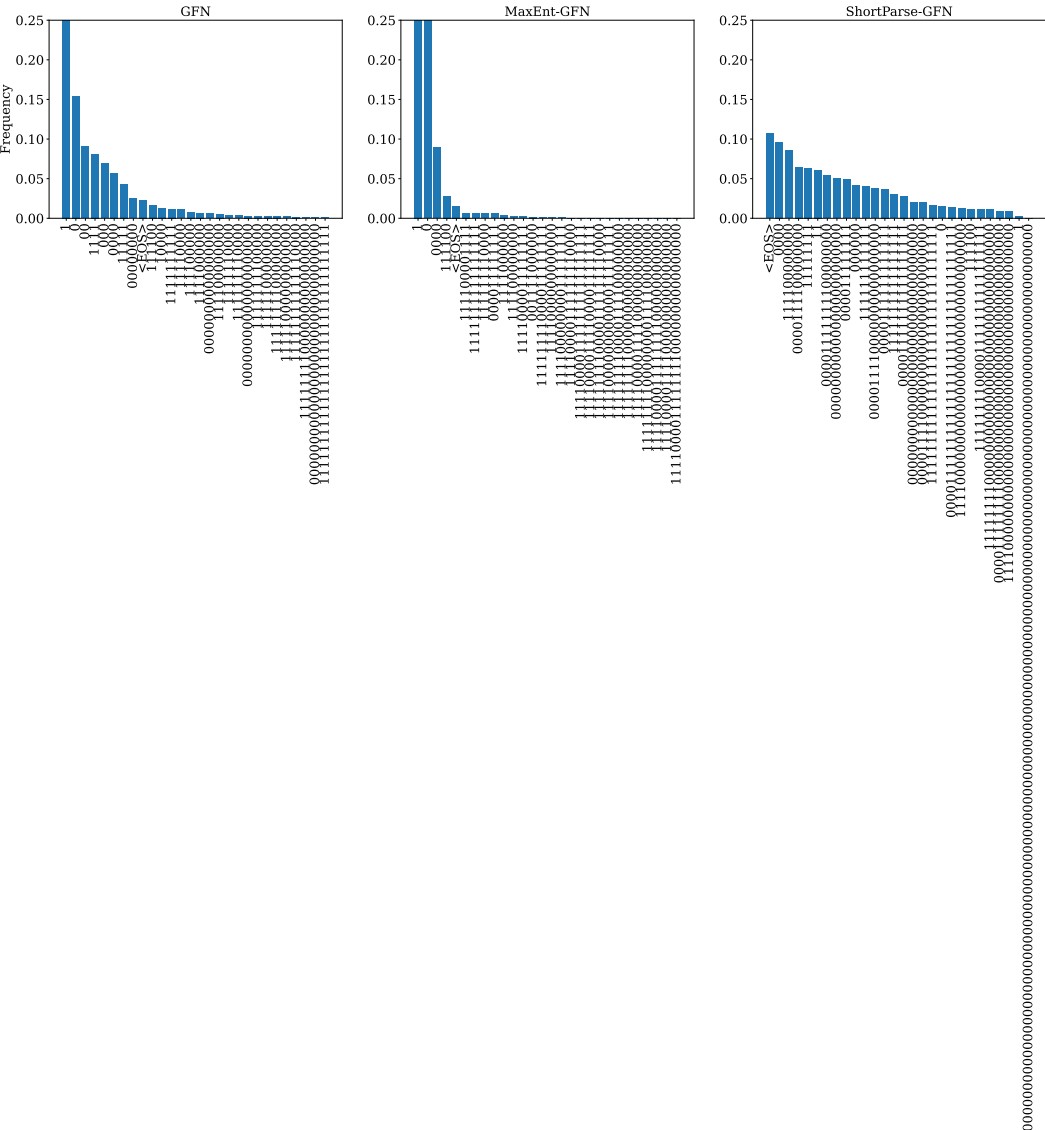

Figure 11: Frequency of use of chunks for the ACTIONPIECE-REPLACE mechanism for GFlowNet in bit sequence for length 128.

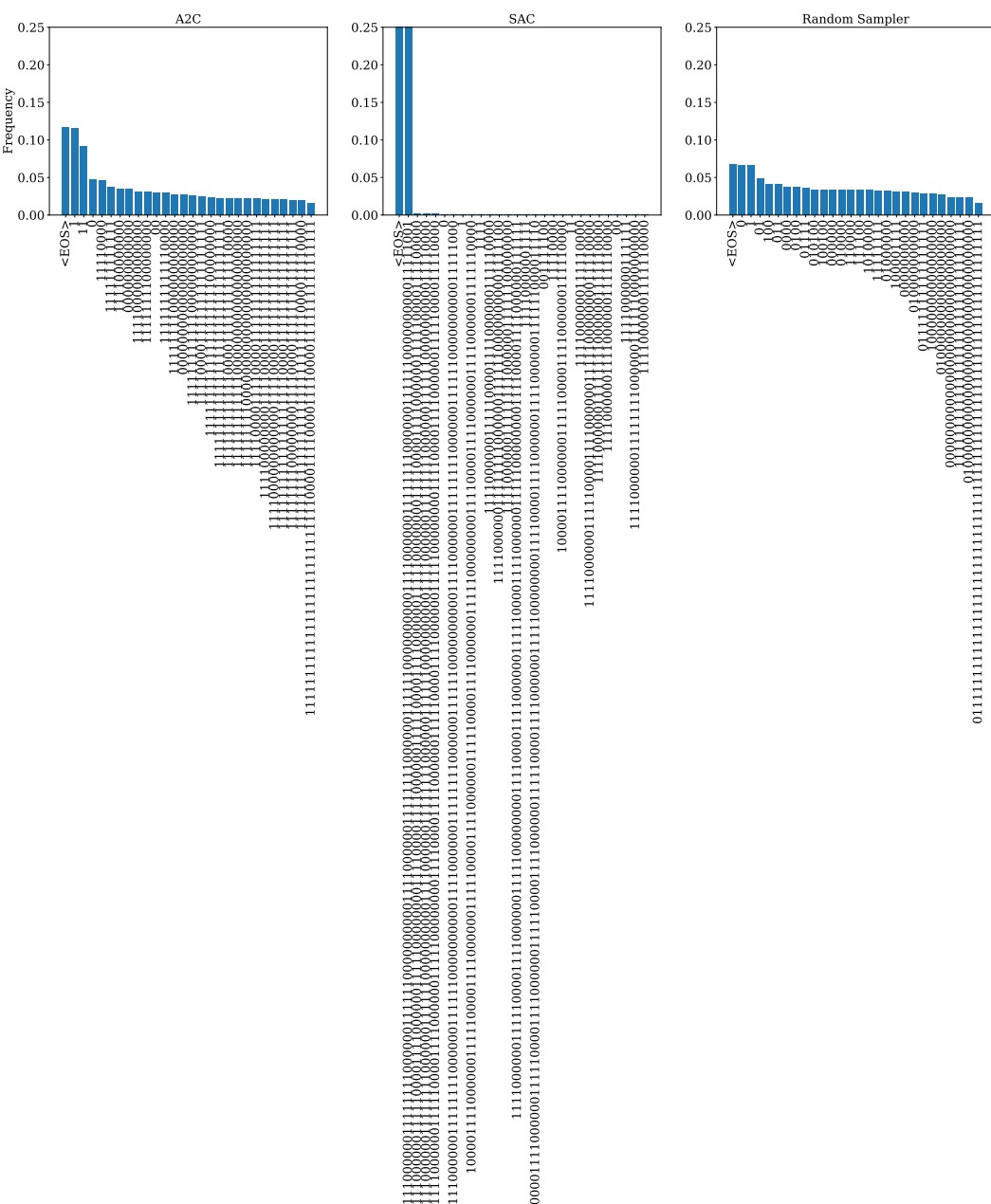

Figure 12: Frequency of use of chunks for the ACTIONPIECE-REPLACE mechanism for RL methods in bit sequence for length 128.

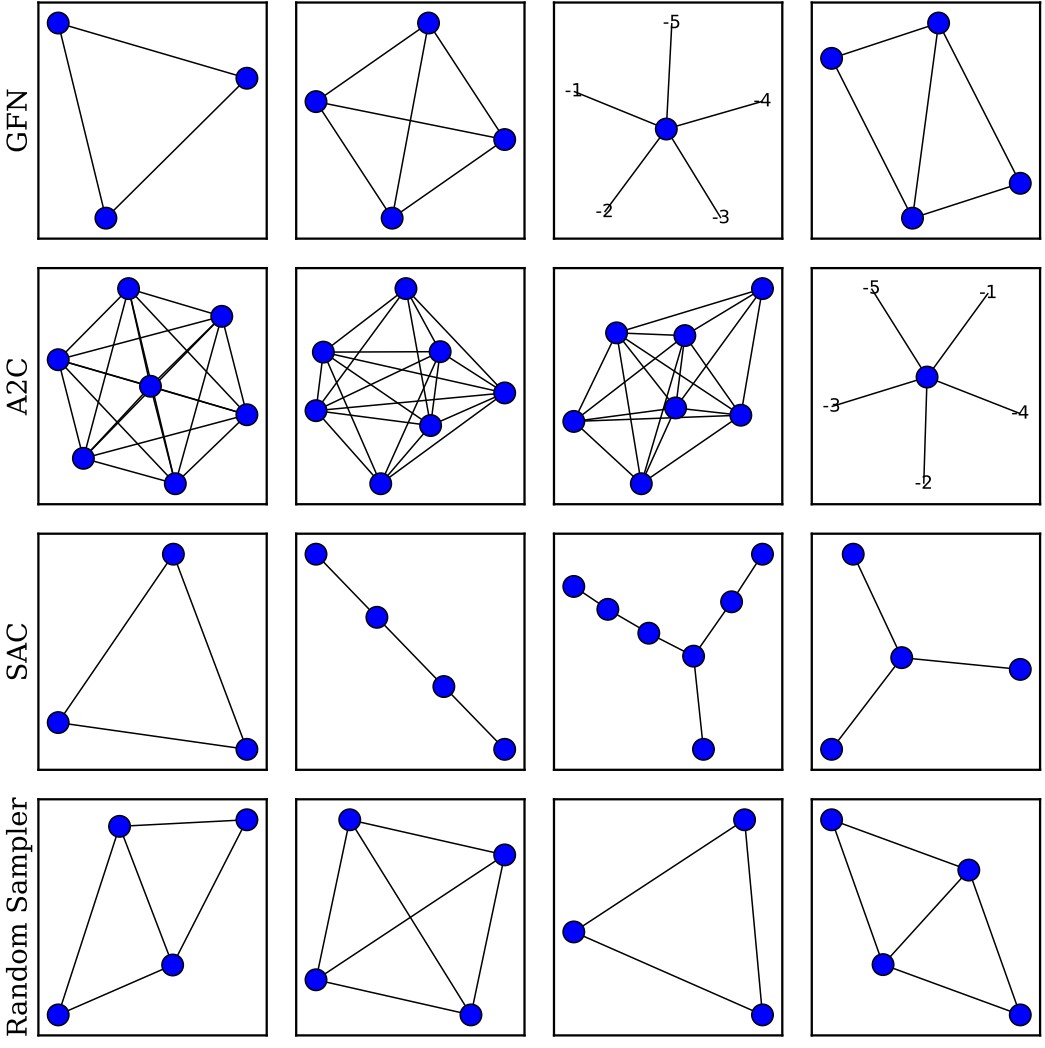

Figure 13: Visualization of a subset of chunks learned by the samplers for the Graph environment with maximum number of nodes of 7. Note for chunks where nodes connected to numbers, these represent the nodes in the graph relative to the last node, that they would be added to.

