# OpenReview forum: "Action abstractions for amortized sampling"
_ICLR.cc/2025/Conference — ICLR 2025 Poster_

### Official Review · Reviewer_juwr · 2024-11-03

**Soundness:** 3
**Presentation:** 3
**Contribution:** 3
**Rating:** 6
**Confidence:** 2

**Summary:**

This paper proposes a method named ACTIONPIECE that incorporates action abstraction discovery into the policy optimization process. It leverages tokenizers to chunk action sequences, which can be viewed as a strategy for trading reduced depth of the MDP in exchange for increased breadth.

**Strengths:**

1. The proposed method is a novel approach to improve the performance of samplers on model discovery and capturing the latent structure of the environment.

2. The method has been validated in three different scenarios, demonstrating its broad applicability and robustness.

**Weaknesses:**

1. The paper lacks rigorous theoretical analysis or proofs to support the observed improvements. While empirical results are strong, a more detailed theoretical discussion would strengthen the claims about the performance of the proposed method.

2. The abstract lacks a quantitative presentation of the results.

3. Can the proposed method be applied to offline-training RL tasks? Can the authors provide some discussion on offline training?

4. As mentioned by the authors, the use of a fixed BPE tokenizer for chunking fixed target distributions in experiments may limit the method's generality and flexibility.

**Questions:**

Please check the weaknesses.

---

> ### Author Response · Authors · 2024-11-20
>
> We thank the reviewer for the helpful comments on our work.
>
> > The paper lacks rigorous theoretical analysis or proofs to support the observed improvements. While empirical results are strong, a more detailed theoretical discussion would strengthen the claims about the performance of the proposed method.
>
> We appreciate the reviewer's concern for theoretical results. However, we decided to limit the scope of our present work to empirical analysis since theoretical analysis on amortized sampling and abstractions is itself limited in prior work.
>
> > Can the proposed method be applied to offline-training RL tasks? Can the authors provide some discussion on offline training?
>
> This is an interesting question, though not quite in the scope of problems we aim to solve with the proposed metohd. We direct you to the response to Reviewer ULj8 regarding [1], which proposes using a two-stage quantized and encoding and BPE procedure to infer macro-actions from an offline continuous control dataset. In essence, the procedure in [1] can be seen as an offline variant of our approach that does not update the token library during training. Combinations of the two approaches could be interesting to explore.
>
> That being said, we will be conducting additional experiments to show results from training using chunks abstracted from an offline dataset for the RNA Binding task.
>
> [1] Zheng, Ruijie, Ching-An Cheng, Hal Daumé III, Furong Huang, and Andrey Kolobov. "PRISE: LLM-Style Sequence Compression for Learning Temporal Action Abstractions in Control." In Forty-first International Conference on Machine Learning.
>
> > As mentioned by the authors, the use of a fixed BPE tokenizer for chunking fixed target distributions in experiments may limit the method's generality and flexibility.
>
> We are not sure to understand the question correctly:
> - Note that the tokenizer is not fixed, as the library of chunks is learned online during training and thus evolves over time. BPE is used as the chunking mechanism, i.e., to decide which new tokens to add to the library.
> - If it was meant that the use of BPE, rather than other compression algorithms, may be limiting -- this is a valid concern, and we performed additional experiments to study the effect of different tokenizers. We kindly direct you to the response to Reviewer A2Lx regarding WordPiece and a uniform tokenizer.
>
> **We would like to thank the reviewer for their helpful comments and would welcome further questions during the discussion period!**

---

> > ### Author Response · Authors · 2024-11-23
> > **Offline dataset results**
> >
> > We conducted additional experiments comparing the use of an offline dataset to build a static library of tokens for samplers versus our dynamic approach. These experiments were performed on two tasks: ``L14_RNA1`` and ``L50_RNA1``, where the latter involves RNA strings with a sequence length of 50, making it significantly more challenging. For both approaches, we built offline datasets by randomly sampling RNA strings and retaining only:
> > * 10,000 RNA strings with Levenshtein distance ≤ 8 for ``L14_RNA1``
> > * 5,000 RNA strings with Levenshtein distance ≤ 15 for ``L50_RNA1``
> >
> > We focused on keeping RNA strings closer to each other rather than farther apart, as this better reflects real-world scenarios where datasets are small and lack diversity. These experiments differ significantly from those in control settings, as amortized sampling typically involves scenarios where large datasets are unavailable for unnormalized probability distributions.
> >
> > For ``L14_RNA1``, we enumerated all RNA strings to determine the modes using a two-step process:
> > 1. Filter strings to retain only those with rewards $\geq 0.9$.
> > 2. Build a set of diverse modes by keeping only RNA strings with a Levenshtein distance of at least $4$ from each other.
> >
> > For ``L50_RNA1``, enumeration is not possible due to the size of the seqrch space ($4^50$). Instead, we tracked all RNA strings with rewards $\geq 0.9$ discovered during training by each sampler. For each sampler, we formed a set of diverse modes by ensuring that the Levenshtein distance between all strings in the set met a given threshold:
> >
> > [Here you can find a link to the results table](https://ibb.co/BGSHrv4).
> >
> > From the results, we observe:
> >
> > ``L14_RNA1``: For GFlowNets, the online discovery of action abstractions is highly competitive with the offline approach. For other samplers, both offline and online discovery of action abstractions perform poorly compared to the baseline of using only atomic actions.
> >
> > ``L50_RNA1``: GFlowNets are competitive in discovering diverse modes (threshold = 15) but slightly underperform the offline dataset at a lower diversity threshold (threshold = 10). For other samplers, neither offline nor online approaches provide any advantage over atomic actions. This limitation may stem from the fact that RL samplers are known to be less exploratory than GFlowNets, causing them to get stuck in local optima induced by the token library.
> >
> > We would like to reiterate that offline datasets for sampling proportionally to an unnormalized distribution are typically not readily available in practice. This is why we focus on analyzing the online version, and our experiments confirm that it is competitive with the offline dataset approach. We also believe that combining both offline and online approaches represents a promising avenue for future research.

---

> > > ### Comment · Reviewer_juwr · 2024-11-27
> > >
> > > Thank you for your response. I must clarify that I am not familiar with the methods proposed by the author and research background, as demonstrated by the confidence 2. Therefore, even though my problem has been solved, I am unable to improve my rating. From the introduction and experimental results in the manuscript, I consider this work is **novel and meaningful**, and the manuscript is well-orginized. I would like to support this work and **recommend acceptance**, ignoring the rating.

---

### Official Review · Reviewer_ULj8 · 2024-11-03

**Soundness:** 2
**Presentation:** 1
**Contribution:** 1
**Rating:** 6
**Confidence:** 4

**Summary:**

The paper investigates the action abstraction for amortized sampling with reinforcement learning and generative flow networks. It proposes to iteratively extract high-reward action subsequences to be expanded as new actions, trading increased breadth for decreased depth of the Markov decision process.  In each iteration, it uses the byte pair encoding for generating high frequency subsequences from the high-reward sequences. Two variants are considered: one incrementally adds new subsequences, while the other periodically replaces old action abstractions.

**Strengths:**

**Quality**: The paper is of good methodological quality. It conducts a careful and meticulous empirical investigation of the use of action abstraction in amortized sampling.

**Significance**: The exploration of action abstraction in this manner could potentially lead to more efficient sampling techniques, although its impact remains to be fully assessed.

**Weaknesses:**

**Novelty**: The use of byte pair encoding for learning action abstraction is not new, as it is already explored in the work by Zheng et al. [1]. To enhance the novelty, the authors should differentiate their approach more clearly or build upon the existing frameworks with significant innovations.

**Significance**: The proposed algorithm exhibits varied performance across different tasks and samplers, lacking a consistent indication of its promise as a robust approach. This raises questions about its generalizability and efficacy in broader applications.

**Clarity**: The conclusions are not effectively communicated. Despite thorough reading, the paper does not present clear, actionable takeaways or insights, as results appear highly case-specific. Improving the clarity of conclusions and providing more generalized insights would increase the paper's impact.

**References**

[1] Zheng, Ruijie, Ching-An Cheng, Hal Daumé III, Furong Huang, and Andrey Kolobov. "PRISE: LLM-Style Sequence Compression for Learning Temporal Action Abstractions in Control." In Forty-first International Conference on Machine Learning.

**Questions:**

- How does your approach specifically differ from the method proposed by Zheng et al. as well as other action abstraction methods?

- The conclusions seem case-specific. Can you provide a more generalized summary of the key takeaways from your research?

---

> ### Author Response · Authors · 2024-11-20
>
> We thank the reviewer for the helpful comments on our work. See below for our comments:
>
> > *Novelty*: The use of byte pair encoding for learning action abstraction is not new, as it is already explored in the work by Zheng et al. [1]. To enhance the novelty, the authors should differentiate their approach more clearly or build upon the existing frameworks with significant innovations.
>
> Thank you for drawing our attention to this recent work. While the use of BPE is similar between the methods, the algorithms and settings are quite different:
>
> In [1], the domain of continuous control is studied. The proposed algorithm proceeds in two stages. First, a quantized encoding function is learned, which maps a pair of (a latent encoding of) an observation and a continuous-valued action to a discrete token. In the second stage, BPE is applied to the sequences of encoded actions to obtain macro-actions. This is very different from our work, which operates in action spaces and learns the library online, i.e., incrementally during training. As a concequence of this, the settings to which these approaches are applicable are very different.
>
> Our work shares with [1] the motivation of extracting high-level, discrete macro-actions and the use of BPE on action sequences to do so. However, there are a number of important differences in the problem setting and approach:
>
> - While [1] studies continuous control, we study amortized sampling in discrete domains, where the set of discrete actions is already available (obviating the need to learn a quantized encoder).
> - On the other hand, while in [1] a dataset of expert trajectories is available, we consider an online setting, where the sampling agent must explore to discover high-reward trajectories.
> - While [1] has a two-stage quantization and compression procedure, we perform discovery of macro-actions online. That is, we iteratively discover new macro-actions in a dynamic dataset of high-reward trajetories discovered so far by the sampler. The growing library of useful action abstractions and the improving amortized sampler reinforce each other in a virtuous loop.
>
> It would be interesting to combine ideas from [1] and from our work. For example, a control agent could be trained in a loop such as the one we propose, in which an agent generates sequences of actions from a dynamically growing library that is initialized with the original quantized actions from [1]'s first stage and iteratively updated by performing BPE on discovered high-reward sequences.
>
> Thank you again for pointing out this work: we now mention its relevance after introducing the algorithm in Section 4.
>
>
> > *Significance*: The proposed algorithm exhibits varied performance across different tasks and samplers, lacking a consistent indication of its promise as a robust approach. This raises questions about its generalizability and efficacy in broader applications.
>
> Thank you for raising issues around the presentation of our results. On consistent finding in this study is that the *utility of macro actions (in the form of ActionPiece)* is dependent on the sampling method used. In our results, the diversity-seeking GFlowNet algorithms made the best use of the chunks learned by ActionPiece: they facilitated the discovery of new modes, learned libraries which provided the shortest parse of the path to the modes in the data, and produced chunks that transferred best to new models or tasks. We have added a section to the conclusions to this effect.
>
> > *Clarity*: The conclusions are not effectively communicated. Despite thorough reading, the paper does not present clear, actionable takeaways or insights, as results appear highly case-specific. Improving the clarity of conclusions and providing more generalized insights would increase the paper's impact. [...]
> > The conclusions seem case-specific. Can you provide a more generalized summary of the key takeaways from your research?
>
> Each section of the paper ends with a short description of the results found. Motivated by your comment, we have expanded the conclusions section of the paper to provide a high-level summary of the impact of this work and the main findings.
>
> [1] Zheng, Ruijie, Ching-An Cheng, Hal Daumé III, Furong Huang, and Andrey Kolobov. "PRISE: LLM-Style Sequence Compression for Learning Temporal Action Abstractions in Control." In Forty-first International Conference on Machine Learning.
>
> **Thank you again for your comments and suggestions, which have helped us improve the paper. We hope that these responses have addressed your concerns and would welcome any further feedback.**

---

> > ### Comment · Reviewer_ULj8 · 2024-11-25
> >
> > The new discussion in the conclusion section stresses the novelty and significance of the combination of GFlowNets and action abstraction, which certainly abates my concerns. Therefore, I have updated my score to 6.

---

> > > ### Author Response · Authors · 2024-11-26
> > > **Thank you for your response**
> > >
> > > We sincerely thank you for taking the time to read our rebuttal. We are pleased that you feel the paper's quality has improved and greatly appreciate your score revision.
> > >
> > > Please feel free to share any additional questions, comments, or suggestions.

---

### Official Review · Reviewer_A2Lx · 2024-11-03

**Soundness:** 3
**Presentation:** 4
**Contribution:** 3
**Rating:** 8
**Confidence:** 3

**Summary:**

This paper addresses the long-planning horizon problem in credit assignment by providing a trunking method ActionPiece. This approach aims to extract high-order actions from sampled trajectories and can be plugged into any sampler. In experiments with three classic algorithms and various environments, the capability of the proposed method in mode discovery, density estimation improvement, and sample length reduction is demonstrated. Abundant discussions and information are further provided.

**Strengths:**

Overall, this paper is well presented and provides an articulate method. I particularly appreciate the environment selections of a real-world orientation and informative way of discussion.

**Weaknesses:**

(W1) Minor typos, e.g.,’the the’ at line 319.

(W2) It seems that all experiments are averaged from only three seeds per line 348 and 507, which is not enough to demonstrate statistical significance in some settings.

**Questions:**

(Q1) As the two proposed chunking mechanisms show contrasting capabilities in multiple aspects, I am curious about the possibility of adaptively combining them. Do you have any related explorations?

(Q2) How does the proposed method perform with other tokenization techniques? Does the chunking method particularly fit some of the tokenization techniques such as BPE used in the paper?

---

> ### Author Response · Authors · 2024-11-20
>
> We thank the reviewer for the helpful comments on our work.
>
> > (W1) Minor typos, e.g.,’the the’ at line 319.
>
>  Thank you for pointing out the typo! It was fixed.
>
> > (W2) It seems that all experiments are averaged from only three seeds per line 348 and 507, which is not enough to demonstrate statistical significance in some settings.
>
> Thank you for this observation. The large number of experiments conducted prevents us from running more seeds for all experiments during the rebuttal period, but we will be sure to do this for the final version.
>
> > (Q1) As the two proposed chunking mechanisms show contrasting capabilities in multiple aspects, I am curious about the possibility of adaptively combining them. Do you have any related explorations?
>
> This is an interesting question, although it brings about a new layer of complexity in the schedule of retokenizations. One possibility would be to use ActionPiece-Increment every $x$ steps and then use ActionPiece-Replace on the learned library every $y$ steps, where $y>x$. Another is to use ActionPiece-Increment, but also to periodically remove macro-actions whose frequency of use falls below an adaptive threshold. Did you have in mind a specific way of combining both mechanisms?
>
> > (Q2) How does the proposed method perform with other tokenization techniques? Does the chunking method particularly fit some of the tokenization techniques such as BPE used in the paper?
>
> In the appendix, we ablated the role of BPE against a *uniform* tokenizer. This tokenizer builds the library online by choosing two tokens at random from the current library and concatenating them to form a new token. We additionally  BPE with WordPiece. Both have been included in the updated manuscript; a direct link to the results is [here](https://ibb.co/Xt30qLs).
>
> We see that BPE and WordPiece significantly outperform the uniform tokenizer, which indicates that tokenizing action sequences is the main driver of the performance. Note that the uniform tokenizer still outperforms the baseline of not introducing any new tokens during training (Atomic), which further reinforces the claim that action abstraction are helpful for training amortized samplers.
>
> WordPiece performs strongly relative to BPE and will be evaluated on all tasks in the final version.
>
> **Thank you for the insightful suggestion, and thank you again for your other comments and questions. We are happy to provide further information if needed.**

---

> > ### Author Response · Authors · 2024-11-20
> > **Continuation of the comment**
> >
> > We also conducted additional experiments using the Option-Critic algorithm, and the results are as follows:
> >
> > Number of modes discovered during training in the `L14_RNA1` environment:
> > | **Sampler**         | **Atomic**     | **ActionPiece-Increment** | **ActionPiece-Replace** |
> > |----------------------|----------------|----------------------------|--------------------------|
> > | GFlowNet             | 43.24 ± 2.21  | 48.67 ± 1.25                    | 41.00 ± 3.27                  |
> > | MaxEnt-GFlowNet      | 43.24 ± 2.21  | 48.33 ± 0.47                    | 42.33 ± 2.87                   |
> > | ShortParse-GFlowNet  | 43.24 ± 2.21  | 48.27 ± 1.32                    | 36.67 ± 3.77                  |
> > | A2C                  | 37.67 ± 0.94  | 24.00 ± 0.82                    | 18.67 ± 3.77                  |
> > | SAC                  | 18.33 ± 2.05  | 0.67 ± 0.47                      | 4.33 ± 2.62                   |
> > | Random Sampler       | 0.00 ± 0.00   | 14.67 ± 2.62                     | 17.67 ± 4.50                  |
> > | Option-Critic       | 16.00 ± 2.16   | ---                  | ---                  |
> >
> > Number of modes discovered during training in the `bit_sequence` environment for sequence length 128:
> >
> > | **Sampler**         | **Atomic**     | **ActionPiece-Increment** | **ActionPiece-Replace** |
> > |---------------------|----------------|---------------------------|-------------------------|
> > | GFlowNet                 | 32.81 (2.0)    | 35.0 (0.27)              | 21.25 (6.23)           |
> > | MaxEnt-GFlowNet          |     32.81 (2.0)      | 39.52 (2.47)              | 35.0 (2.94)             |
> > | ShortParse-GFlowNet      |  32.81 (2.0)   | 36.09 (1.64)             | 12.33 (1.25)           |
> > | A2C                 | 9.67 (7.76)    | 0.0 (0.0)                | 26.33 (18.66)          |
> > | SAC                 | 0.0 (0.0)      | 0.0 (0.0)                | 0.0 (0.0)              |
> > | Random Sampler      | 0.0 (0.0)      | 0.0 (0.0)                | 4.0 (0.82)             |
> > | Option-Critic       | 0.67 (0.94)    |---|---|
> >
> > Number of modes discovered during training for the ``FractalGrid`` for size 257:
> >
> > | **Sampler**         | **Atomic**     | **ActionPiece-Increment** | **ActionPiece-Replace** |
> > |---------------------|----------------|---------------------------|-------------------------|
> > | GFN                 | 1.0 ± 0.0      | 16.0 ± 0.0                | 16.0 ± 0.0              |
> > | A2C                 | 1.0 ± 0.0      | 3.67 ± 2.36               | 16.0 ± 0.0              |
> > | SAC                 | 1.0 ± 0.0      | 2.33 ± 1.89               | 8.33 ± 0.94             |
> > | Random Sampler      | 1.0 ± 0.0      | 10.33 ± 0.47              | 15.67 ± 0.47            |
> > | Option-Critic       | 1.0 ± 0.0      |---|---|
> >
> > Note that Option-Critic should be compared with the ActionPiece-Increment/Replace columns of the other samplers since both learn macro-actions. We can see that the Option-Critic sampler struggles.
> >
> > **We hope that these results results have satisfied the reviewer and would welcome further feedback before the end of the discussion period.**

---

> > ### Comment · Reviewer_A2Lx · 2024-11-26
> >
> > Thank you for your efforts in addressing my questions and delivering additional experiments. I will keep an eye on your final version regarding W2 (: Good Luck!

---

> > > ### Author Response · Authors · 2024-11-28
> > > **3 Additional Seed Ran**
> > >
> > > Hello - we have dedicated significant compute to produce 3 new seeds for the experiments in Figure 2, Table 2, and Figure 4. In addition, we have explicitly split out the seeds in Figure 4 so one can more easily visualize the size of the average shortest parse across seeds in each condition. We hope this gives the reader a clearer view of the variance of each method across each seed. These new results have been uploaded just now, with no changes to the interpretation results in our estimation.

---

### Official Review · Reviewer_QHVg · 2024-11-03

**Soundness:** 3
**Presentation:** 4
**Contribution:** 3
**Rating:** 8
**Confidence:** 2

**Summary:**

The paper proposes a new technique to iteratively extract action subsequences from high-reward trajectories as temporally extended skills that are then added to the action space for more efficient exploration. Applying this technique on top of generative flow networks (GFlowNets) yields improved sample efficiency in hard exploration problems. The authors experiment on three synthetic tasks and an RNA sequence generation task and show that the proposed method improves GFlowNets and outperform prior methods (e.g., RL methods like A2C and SAC) on these tasks.

**Strengths:**

- The idea of expanding the action space dynamically online with temporally extended action sequence is interesting and novel.
- The experiments are comprehensive and thorough with insightful analyses and visualizations that demonstrate the effectiveness of the proposed algorithm.

**Weaknesses:**

*Unfounded claim*
- “the abstracted high-level actions are interpretable, …” — there is no evidence presented in the paper that illustrates the high-level actions are interpretable.

*Comparison to prior chunking mechanisms is limited*
- The authors considered two new chunking mechanisms, "ActionPiece-Increment" and "ActionPiece-Replace". Both of them use heuristics to expand action space with temporally extended action sequences.
- It is unclear how these mechanisms compare to prior chunking mechanisms used in the options/unsupervised skill discovery/hierarchical RL literature (e.g., with variational formualtion [1], with clustering [2].

[1] Kim, Taesup, Sungjin Ahn, and Yoshua Bengio. "Variational temporal abstraction." Advances in Neural Information Processing Systems 32 (2019).

[2] Srinivas, Aravind, et al. "Option discovery in hierarchical reinforcement learning using spatio-temporal clustering." arXiv preprint arXiv:1605.05359 (2016).

**Questions:**

N/A

---

> ### Author Response · Authors · 2024-11-20
>
> We thank the reviewer for helpful comments on our work.
>
> > “the abstracted high-level actions are interpretable, …” — there is no evidence presented in the paper that illustrates the high-level actions are interpretable.
>
> In Section 6.2, we answer the interpretabilty question indirectly by showing, using various probes, that the learned macro-actions capture some underlying structure in the target distribution. While interpretabiilty is difficult to measure, in Appendix C we show the libraries of learned chunks for all tasks, which exhibit clear patterns consistent with the structure of the environment (see, e.g., discussion in Appendix C.1). We are happy to point to these results more prominently in the main text.
>
> > The authors considered two new chunking mechanisms, "ActionPiece-Increment" and "ActionPiece-Replace". Both of them use heuristics to expand action space with temporally extended action sequences.
> It is unclear how these mechanisms compare to prior chunking mechanisms used in the options/unsupervised skill discovery/hierarchical RL literature (e.g., with variational formualtion [1], with clustering [2].
>
> We thank the reviewer for pointing out specific examples in the literature:
>
> - In [1], the authors learn a generative model that decomposes a sequence of observations into non-overlapping subsequences. They assume the existence of latent temporal abstractions that generate these subsequences and consider the number and length of each subsequence as latent variables, rather than treating them as mere hyperparameters. To achieve this, they introduce a "Binary Subsequence Indicator" variable to switch between temporal abstractions, akin to the termination probability in Options. However, their approach relies on the availability of a dataset and the overhead cost of training the generative model. In contrast, our method directly extracts the most common subsequences online at a negligible computational cost.
> - In [2], authors cluster states of the MDP to extract abstract states (macrostates) and they learn options between these macrostates. While their method is similar in that it is conducted online, our approach operates directly in the action space, making it applicable to any environment. Their method, on the other hand, depends on the quality of state embeddings. Additionally, our approach incurs negligible computational cost, whereas theirs requires training the newly mined options.
>
> In general, our method differs from existing literature by focusing on abstractions in the context of amortized sampling. We target environments where high-reward states exhibit clear repeating substructures. Moreover, we extract abstractions online using BPE, which incurs negligible cost and facilitates quick adaptation during training, thanks to our action encoder that can generate embeddings for any sequence of actions making up a macro-action, even one not previously seen.

---

> > ### Comment · Reviewer_QHVg · 2024-11-26
> >
> > Thanks for addressing the concerns!
> >
> > For interpretability point, I understand that you showed the abstracted high-level actions have captured structures in the environment, but that alone does not make the actions interpretable. Interpretable actions should have the characteristics where the purpose/consequence of the action can be clearly explained. Without that, I would still recommend not using the word "interpretable" to describe your high-level actions because there is no clear metric in the paper to show how interpretable the high-level actions are. Perhaps a better way of describing it is to directly use more descriptive/informative phrases like what you mentioned -- "the learned macro-actions capture some underlying structure..".
> >
> > In addition, in the conclusion paragraph the paper mentions that "the learned chunks are potentially interpretable", but in the abstract, the tone seems much stronger: "We also observe that the abstracted high-order actions are interpretable". I would recommend to make them more consistent with each other.

---

> > > ### Author Response · Authors · 2024-11-27
> > > **RE Interpretability**
> > >
> > > Thank you for this suggestion. We agree with your assessment and have opted to soften the language "potentially interpretable", and "learned chunks capture the underlying structure of the reward distribution (and are therefore potentially interpretable)". We believe the interpretability of the learned chunks can be greatly improved in future work were the chunking method is explicitly design to produce chunks with a human-friendly structure - we leave that to future work and make mention of this in the appendix.

---

### Official Review · Reviewer_3r38 · 2024-11-04

**Soundness:** 3
**Presentation:** 2
**Contribution:** 2
**Rating:** 6
**Confidence:** 4

**Summary:**

This paper introduces a approach to incorporate action abstractions into the policy optimization process of reinforcement learning and generative flow networks. The method, termed ACTIONPIECE, aims to improve sample efficiency and mode discovery by iteratively extracting and chunking action subsequences from high-reward trajectories into a growing action space. The experimental results demonstrate the effectiveness of the proposed method.

**Strengths:**

- The proposed ACTIONPIECE compatible with both RL and GFlowNets sampler.
- Empirical evaluation showing improved sample efficiency and mode discovery in different environments.

**Weaknesses:**

- As mentioned in the related works section, the discovery of macro-actions has been extensively studied. The authors should provide a more detailed discussion highlighting how the proposed method differs from existing methods.

- In Line 159, the paper appears to assume a deterministic state transition, where $s'=s+a$. This assumption may be too strong and not applicable to real-world environments where state transitions involve a degree of randomness. The reviewers are concerned about the generalizability of the proposed method to stochastic environments. The authors should address whether and how the method can accommodate stochastic state transitions, which are common in many practical applications of RL and GFlowNets.

- The reviewers suggest that the authors should provide a more comprehensive introduction to the concept of "amortized samplers" in the Preliminaries section. It is essential for readers who are unfamiliar with this concept.

- The proposed Algorithm in Section 4 seems to be applicable to both GFlowNets and RL methods with discrete action spaces, as evidenced by the experiments conducted in the paper. However, the authors have chosen to focus heavily on GFlowNets as the primary background, which may not be immediately clear to readers. The reviewers recommend that the authors clarify why GFlowNets were chosen as the main framework and how the proposed method specifically leverages or addresses challenges unique to GFlowNets.

- In Line 304, the paper mentions an action encoder but does not elaborate on how it is trained. The authors should provide details on the training process of the action encoder.

- The experiments presented in the paper is simple. The reviewers question why the authors did not test their method in more complex environments, such as Atari games, which are known for their high dimensionality and complexity. Additionally, the paper lacks a comparison with existing methods for discovering macro-actions.

**Questions:**

See Weaknesses.

---

> ### Author Response · Authors · 2024-11-20
>
> We thank the reviewer for their helpful comments on our work.
> > As mentioned in the related works section, the discovery of macro-actions has been extensively studied. The authors should provide a more detailed discussion highlighting how the proposed method differs from existing methods.
>
> Our work is different from existing literature in the following aspects:
> - We study the effect of action abstractions on the training of amortized samplers, which, to the best of our knowledge, has not been explored before.
> - We investigate the discovery of macro-actions online during training and dynamically modify the action space. This means the action space is no longer static but evolves throughout the training process. Known frameworks for temporal abstraction, such as Options, focus on discovering policies while keeping the action space constant. Furthermore, these frameworks are closed-loop, whereas ours operates in an open-loop manner.
> - We deliberately focus on environments where high-reward samples exhibit clear, repeating substructures, as opposed to complex environments (e.g., Atari) with entangled generalization factors that would make it challenging to derive clear and actionable conclusions.
> > In Line 159, the paper appears to assume a deterministic state transition, where $𝑠'=𝑠+𝑎$. This assumption may be too strong and not applicable to real-world environments where state transitions involve a degree of randomness. The reviewers are concerned about the generalizability of the proposed method to stochastic environments. The authors should address whether and how the method can accommodate stochastic state transitions, which are common in many practical applications of RL and GFlowNets.
>
> In this paper, we explore the use of action abstractions specifically for *amortized sampling* -- generation of samples from an unnormalized target density via a hierarchical generative process (i.e., by sequential decision-making). In such settings, the MDP is assumed to be deterministic (as is, in fact, done in all existing applications of GFlowNets for amortized sampling).
>
> Generalization to sampling settings with stochastic environments -- while it would not constitute amortized inference in the commonly understood sense -- would be a direction for future work. One interesting case is action abstractions in adversarial games, where generalizations of GFlowNet objectives have been developed [Vucetic et al., "Expected flow networks...", ICLR'24]. We note that macro-action discovery in multi-agent or adversarial settings is largely unexplored in traditional RL domains as well.
>
> > The reviewers suggest that the authors should provide a more comprehensive introduction to the concept of "amortized samplers" in the Preliminaries section. It is essential for readers who are unfamiliar with this concept.
>
> We have added the following to section 3 of the manuscript (**Preliminaries**): *Amortized sampling is the process of sampling from a functional approximation of the target distribution, where the computational cost of iterative sampling is shifted to the model's optimization process, enabling rapid sampling after training is complete.*
>
> > The proposed Algorithm in Section 4 seems to be applicable to both GFlowNets and RL methods with discrete action spaces, as evidenced by the experiments conducted in the paper. However, the authors have chosen to focus heavily on GFlowNets as the primary background, which may not be immediately clear to readers. The reviewers recommend that the authors clarify why GFlowNets were chosen as the main framework and how the proposed method specifically leverages or addresses challenges unique to GFlowNets.
>
> GFlowNets enable sampling from an unnormalized distribution, and our focus is specifically on their role in studying the impact of abstractions on generative modeling. We selected environments frequently used in GFlowNet research, as they provide a controlled setting for analyzing how abstractions influence generative modeling.
>
> > In Line 304, the paper mentions an action encoder but does not elaborate on how it is trained. The authors should provide details on the training process of the action encoder.
>
> The purpose of the action encoder is simply to provide embeddings for macro-actions. There is no separate loss to train the action encoder: for both GFlowNets and RL baselines, the policy logits are computed using Equation (4), so when the policy loss is backpropagated, it changes the parameters of the action encoder as well.

---

> > ### Author Response · Authors · 2024-11-20
> > **Continuation of the comment**
> >
> > > The experiments presented in the paper is simple. The reviewers question why the authors did not test their method in more complex environments, such as Atari games, which are known for their high dimensionality and complexity.
> >
> > Our work is focused on investigating the effect of macro-actions for the problem of *sampling* from an distribution given its unnormalized density. Classical RL environments, such as Atari, do not fit this scope.
> >
> > We note, however, that our approach draws inspiration from action abstraction methods in RL. Conversely, we hope and expect that our work can inspire methods applicable to RL and control (see, e.g., the response to Reviewer ULj8 regarding PRISE).
> >
> > > Additionally, the paper lacks a comparison with existing methods for discovering macro-actions.
> >
> > To the best of our knowledge, our work is the first to study macro-actions in the context of sampling. We have run additional experiments using the Option-Critic algorithm and got the following results:
> >
> > Number of modes discovered during training in the `L14_RNA1` environment:
> > | **Sampler**         | **Atomic**     | **ActionPiece-Increment** | **ActionPiece-Replace** |
> > |----------------------|----------------|----------------------------|--------------------------|
> > | GFlowNet             | 43.24 ± 2.21  | 48.67 ± 1.25                    | 41.00 ± 3.27                  |
> > | MaxEnt-GFlowNet      | 43.24 ± 2.21  | 48.33 ± 0.47                    | 42.33 ± 2.87                   |
> > | ShortParse-GFlowNet  | 43.24 ± 2.21  | 48.27 ± 1.32                    | 36.67 ± 3.77                  |
> > | A2C                  | 37.67 ± 0.94  | 24.00 ± 0.82                    | 18.67 ± 3.77                  |
> > | SAC                  | 18.33 ± 2.05  | 0.67 ± 0.47                      | 4.33 ± 2.62                   |
> > | Random Sampler       | 0.00 ± 0.00   | 14.67 ± 2.62                     | 17.67 ± 4.50                  |
> > | Option-Critic       | 16.00 ± 2.16   | ---                  | ---                  |
> >
> > Number of modes discovered during training in the `bit_sequence` environment for sequence length 128:
> >
> > | **Sampler**         | **Atomic**     | **ActionPiece-Increment** | **ActionPiece-Replace** |
> > |---------------------|----------------|---------------------------|-------------------------|
> > | GFlowNet                 | 32.81 (2.0)    | 35.0 (0.27)              | 21.25 (6.23)           |
> > | MaxEnt-GFlowNet          |     32.81 (2.0)      | 39.52 (2.47)              | 35.0 (2.94)             |
> > | ShortParse-GFlowNet      |  32.81 (2.0)   | 36.09 (1.64)             | 12.33 (1.25)           |
> > | A2C                 | 9.67 (7.76)    | 0.0 (0.0)                | 26.33 (18.66)          |
> > | SAC                 | 0.0 (0.0)      | 0.0 (0.0)                | 0.0 (0.0)              |
> > | Random Sampler      | 0.0 (0.0)      | 0.0 (0.0)                | 4.0 (0.82)             |
> > | Option-Critic       | 0.67 (0.94)    |---|---|
> >
> > Number of modes discovered during training for the ``FractalGrid`` for size 257:
> >
> > | **Sampler**         | **Atomic**     | **ActionPiece-Increment** | **ActionPiece-Replace** |
> > |---------------------|----------------|---------------------------|-------------------------|
> > | GFN                 | 1.0 ± 0.0      | 16.0 ± 0.0                | 16.0 ± 0.0              |
> > | A2C                 | 1.0 ± 0.0      | 3.67 ± 2.36               | 16.0 ± 0.0              |
> > | SAC                 | 1.0 ± 0.0      | 2.33 ± 1.89               | 8.33 ± 0.94             |
> > | Random Sampler      | 1.0 ± 0.0      | 10.33 ± 0.47              | 15.67 ± 0.47            |
> > | Option-Critic       | 1.0 ± 0.0      |---|---|
> >
> > Note that Option-Critic should be compared with the ActionPiece-Increment/Replace columns of the other samplers since both learn macro-actions. We can see that the Option-Critic sampler struggles.
> >
> > **Thank you again for the helpful comments, and we hope that these responses and additional experiments have addressed your concerns.**

---

> > > ### Comment · Reviewer_3r38 · 2024-11-24
> > > **Thanks for your response**
> > >
> > > Thanks for your response. I have updated my score. Good luck!

---

> > > > ### Author Response · Authors · 2024-11-26
> > > > **Thank your response**
> > > >
> > > > We sincerely thank you for taking the time to read our rebuttal. We are pleased that you feel the paper's quality has improved and greatly appreciate your score revision.
> > > >
> > > > Please feel free to share any additional questions, comments, or suggestions.

---

### Author Response · Authors · 2024-11-20
**Response to all reviewers + new experiments**

We thank the reviewers for their efforts in reviewing our paper. The comments have helped us to improve the work. We answer each reviewer separately, but use this comment to clarify some recurring points that the reviewers raised and new experiments we have added. Additionally, we highlighted the parts that we added to the manuscript in blue.

**Novelty and contribution**

Our aim is to study the effect of discovering action abstractions in the context of training an amortized sampler. As a consequence, we focus on environments that clearly have repeating substructures, rather than complex RL environments with entangled generalization factors. We believe that our work is the first to study action abstractions in the context of amortized sampling, and it gives actionable insights on how to improve the training of policies for amortized inference over compositional structures.

Our approach draws inspiration from action abstraction methods in RL. Conversely, we hope and expect that our work can inspire methods applicable to RL and control.

**New experiments**

- **Comparison to other macro-action frameworks**: We trained an Option-Critic [Bacon et al., The Option-Critic architecture] agent on all environments and summarized our findings in the following table:

Number of modes discovered during training in the `L14_RNA1` environment:
| **Sampler**         | **Atomic**     | **ActionPiece-Increment** | **ActionPiece-Replace** |
|----------------------|----------------|----------------------------|--------------------------|
| GFlowNet             | 43.24 ± 2.21  | 48.67 ± 1.25                    | 41.00 ± 3.27                  |
| MaxEnt-GFlowNet      | 43.24 ± 2.21  | 48.33 ± 0.47                    | 42.33 ± 2.87                   |
| ShortParse-GFlowNet  | 43.24 ± 2.21  | 48.27 ± 1.32                    | 36.67 ± 3.77                  |
| A2C                  | 37.67 ± 0.94  | 24.00 ± 0.82                    | 18.67 ± 3.77                  |
| SAC                  | 18.33 ± 2.05  | 0.67 ± 0.47                      | 4.33 ± 2.62                   |
| Random Sampler       | 0.00 ± 0.00   | 14.67 ± 2.62                     | 17.67 ± 4.50                  |
| Option-Critic       | 16.00 ± 2.16   | ---                  | ---                  |

Number of modes discovered during training in the `bit_sequence` environment for sequence length 128:

| **Sampler**         | **Atomic**     | **ActionPiece-Increment** | **ActionPiece-Replace** |
|---------------------|----------------|---------------------------|-------------------------|
| GFlowNet                 | 32.81 (2.0)    | 35.0 (0.27)              | 21.25 (6.23)           |
| MaxEnt-GFlowNet          |     32.81 (2.0)      | 39.52 (2.47)              | 35.0 (2.94)             |
| ShortParse-GFlowNet      |  32.81 (2.0)   | 36.09 (1.64)             | 12.33 (1.25)           |
| A2C                 | 9.67 (7.76)    | 0.0 (0.0)                | 26.33 (18.66)          |
| SAC                 | 0.0 (0.0)      | 0.0 (0.0)                | 0.0 (0.0)              |
| Random Sampler      | 0.0 (0.0)      | 0.0 (0.0)                | 4.0 (0.82)             |
| Option-Critic       | 0.67 (0.94)    |---|---|

Number of modes discovered during training for the ``FractalGrid`` for size 257:

| **Sampler**         | **Atomic**     | **ActionPiece-Increment** | **ActionPiece-Replace** |
|---------------------|----------------|---------------------------|-------------------------|
| GFN                 | 1.0 ± 0.0      | 16.0 ± 0.0                | 16.0 ± 0.0              |
| A2C                 | 1.0 ± 0.0      | 3.67 ± 2.36               | 16.0 ± 0.0              |
| SAC                 | 1.0 ± 0.0      | 2.33 ± 1.89               | 8.33 ± 0.94             |
| Random Sampler      | 1.0 ± 0.0      | 10.33 ± 0.47              | 15.67 ± 0.47            |
| Option-Critic       | 1.0 ± 0.0      |---|---|

Note that Option-Critic should be compared with the ActionPiece-Increment/Replace columns of the other samplers since both learn macro-actions. We can see that the Option-Critic sampler struggles.

- **Comparison to WordPiece**: We studied action abstraction when replacing BPE with WordPiece and updated Table 3 from the appendix in the manuscript. We also provide a screenshot of the updated table [here](https://ibb.co/Xt30qLs). We observe that WordPiece results in better performance than BPE with all the samplers for both ActionPiece-Increment and ActionPiece-Replace. Thank you to Reviewer A2Lx for this insightful suggestion!

We appreciate greatly these comments, which believe have helped us strengthen the paper.

---

### Author Response · Authors · 2024-11-28
**Final thank you for all reviewers.**

We would like to take a moment to thank all of the reviewers for their helpful comments which we believe have dramatically improved our paper. In this latest revision which we just uploaded, we have introduced the results from doubling the number of seeds for the core set of experiments, improving the communication of our results, and improving the visualization of Figure 4. All meaningful changes to the text are highlighted in blue. and we look forward to addressing any further comments or concerns you have through the remainder of the discussion period.

---

### Meta-Review · Area_Chair_iiBa · 2024-12-19

**Metareview:**

This paper proposes ACTIONPIECE, a novel approach for incorporating action abstraction discovery into policy optimization for amortized sampling, using tokenizers to chunk action sequences from high-reward trajectories. While reviewers appreciated the paper's methodological rigor and comprehensive empirical validation across different environments, particularly the effectiveness demonstrated with GFlowNets, there were some concerns about the theoretical foundations and generalizability of the approach. The authors provided thorough responses, including additional experiments with Option-Critic baselines and offline datasets, which helped clarify the method's advantages in the amortized sampling setting. The revisions also improved clarity regarding the novelty relative to recent work like PRISE and strengthened the presentation of key findings. Notably, the results consistently show that GFlowNet-based methods make better use of the learned action abstractions compared to other sampling approaches.

Given the paper's strong empirical contributions, clear exposition, and the authors' comprehensive response to reviewer concerns, I recommend acceptance.

**Additional Comments On Reviewer Discussion:**

While reviewers appreciated the paper's methodological rigor and comprehensive empirical validation across different environments, particularly the effectiveness demonstrated with GFlowNets, there were some concerns about the theoretical foundations and generalizability of the approach. The authors provided thorough responses, including additional experiments with Option-Critic baselines and offline datasets, which helped clarify the method's advantages in the amortized sampling setting. The revisions also improved clarity regarding the novelty relative to recent work like PRISE and strengthened the presentation of key findings.

---

### Decision · Program_Chairs · 2025-01-22

Accept (Poster)